# Comments on Two Controversial Oriental Assassin Bug Species of the Genus *Rhynocoris* (Heteroptera: Reduviidae: Harpactorinae), with the Description of *R. minutus* sp. nov. from China [note 1]

**DOI:** 10.3390/insects16080823

**Published:** 2025-08-08

**Authors:** Huaiyu Liu, Zhuo Chen, Haoyang Xiong, Zhaoyang Chen, Hu Li, Ping Zhao, Wanzhi Cai

**Affiliations:** 1State Key Laboratory of Agricultural and Forestry Biosecurity, MOA Key Lab of Pest Monitoring and Green Management, College of Plant Protection, China Agricultural University, Beijing 100193, China; huaiyuliu0103@163.com (H.L.); insectchen625@126.com (Z.C.); cqxiong_hy@126.com (H.X.); zhaoyangchen@cau.edu.cn (Z.C.); tigerleecau@hotmail.com (H.L.); 2Key Laboratory of Environment Change and Resources Use in Beibu Gulf, Ministry of Education, Nanning Normal University, Nanning 530001, China

**Keywords:** Harpactorinae, *Rh*
*ynocoris*, taxonomy, key, new species, China

## Abstract

The genus *Rhynocoris* Hahn, 1834 is a group of medium-sized and brightly colored insects in the reduviid subfamily Harpactorinae, and has proved to play a significant role in the control of agricultural pests. Based on the examination of type specimens, two *Rhynocoris* species widely distributed in the Oriental Region, which exhibit high morphological similarity, were redescribed and clearly distinguished, and their biological information was recorded. Additionally, one new species, *Rhynocoris minutus* Liu, Zhao & Cai, **sp. nov.** was discovered in China.

## 1. Introduction

The assassin bug genus *Rhynocoris* Hahn, 1834 (Hemiptera: Heteroptera: Reduviidae: Harpactorinae) is a species-rich group of about 150 described species distributed mainly in the Eastern Hemisphere, with only two species occurring in North America [1]. The majority of species are distributed in the Ethiopian Region (>80 spp.), and others are distributed in the Oriental and Palearctic regions [1,2]. *Rhynocoris* can be recognized within the Harpactorinae by the following morphological characters: the unarmed head and pronotum; the anterior pronotal lobe is subequal to or longer than half of the posterior lobe in length; and the short median longitudinal sulcus of the anterior pronotal lobe, which reaches neither the collar anteriorly nor the transverse constriction posteriorly [3,4]. Adults of *Rhynocoris* are diurnal and can be found in various vegetated habitats, feeding on insects and other small arthropods. Some species of this genus, such as *R. kumarii* Ambrose and Livingstone, 1986, *R. marginatus* (Fabricius, 1781), and *R. segmentarius* (Germar, 1837), are frequently found in farmlands and exhibit a broad prey range, which have potential as natural enemies for use in biological control programs [5,6,7].

Twelve species of *Rhynocoris* have been recorded in China prior to this study [1,2,3,8,9,10,11]. However, all these species were previously arranged under the generic name *Harpactor* Laporte, 1833 in the previous literature until the publication of the catalogues of Maldonado-Capriles, and Putshkov and Putshkov [1,2]. Due to misidentification caused by the morphological similarity of *Rhynocoris costalis* (Stål, 1867) and *Rhynocoris fuscipes* (Fabricius, 1787) and their importance as natural enemies of agricultural pests in China [3,12,13,14], we conducted a reexamination of the type specimens and redescribed these two taxonomically controversial species. Furthermore, the biological information of these two species was reported to provide further insights into the important group of reduviids. Additionally, we discovered a new species, *R. minutus* Liu, Zhao & Cai sp. nov., which has similar color pattern but is smaller in size. A key is provided for the identification of the *Rhynocoris* species occurring in China.

## 2. Materials and Methods

### 2.1. Specimens and Acronyms

Specimens examined or cited in this study are deposited in the following institutions:

**CAU** Entomological Museum, China Agricultural University, Beijing, China**ZMUC** Zoological Museum of University of Copenhagen, Copenhagen, Denmark**NHMUK** Natural History Museum, London, United Kingdom**NHRS** Swedish Museum of Natural History, Stockholm, Sweden**RBINS** Royal Belgian Institute of Natural Sciences, Brussels, Belgium

### 2.2. Taxonomy

Male genitalia were soaked in hot 10% NaOH solution for approximately five minutes to remove soft tissue, rinsed in distilled water, and dissected under a Motic binocular dissecting microscope. After examination, each dissected genitalia was placed in a vial with glycerin and pinned under the corresponding specimen. Photographs were taken using a Canon EOS R7 digital camera with Canon macro lens EF 100 mm and MP-E 65 mm (Canon Inc., Tokyo, Japan) for habitus, and an Olympus BX51 microscope (Olympus Inc., Tokyo, Japan) for dissected genitalia. Helicon Focus 8.1.0 (Helicon Soft Ltd., Kiev, Ukraine) was used for image stacking. Measurements were obtained using a calibrated micrometer. All measurements are given in millimeters. Morphological terminology mainly followed Davis [15], Hsiao and Ren [3], and Weirauch [16].

### 2.3. Biological Study

The live samples of *Rhynocoris costalis* (Stål, 1867) were collected from Shaoguan city, Guangdong province, China. The samples of *Rhynocoris fuscipes* (Fabricius, 1787) were from Buga town, Zhaotong city, Yunnan province, China. These two species are common natural enemies in pest control in South China. We collected these specimens by hand in agricultural fields. The reduviids were all reared under laboratory conditions (26 ± 2 °C, natural light) and fed on the larvae of the yellow mealworm, *Tenebrio molitor* (Linnaeus, 1758) (Coleoptera: Tenebrionidae). The eggs and 1–5 instars nymphs were described and photographed on artificially placed leaves in a laboratory setting.

## 3. Results

### 3.1. Systematics


**Genus *Rhynocoris* Hahn, 1834**
*Rhynocoris* Hahn, 1834: 20 [17]. **Type species** by subsequent designation (Kirkaldy, 1900: 242 [18]): *Reduvius cruentus* Fabricius, 1787 (= *Cimex iracundus* Poda, 1761).*Harpactor* (non Laporte, 1833): Amyot and Serville, 1843: 364, part [19]; Hsiao and Ren, 1981: 529 [3].*Rhinocoris* Kolenati, 1857: 460 [20]. Unjustified emendation.*Oncauchenius* Stål, 1872: 46 [21]. Type species by subsequent designation (Villiers, 1948: 55 [22]): *Cimex annulatus* Linnaeus, 1758. As subgenus of *Reduvius*.*Chirillus* Stål, 1874: 38 [23]. Type species by subsequent designation (Jeannel, 1919: 290 [24]): *Reduvius marginatus* Fabricius, 1781. As subgenus of *Reduvius*.*Harpiscus* Stål, 1874: 39 [23]. Type species by subsequent designation (Jeannel, 1919: 287 [24]): *Harpactor tropicus* Herrich-Schaeffer, 1848. As subgenus of *Reduvius*.*Lamphrius* Stål, 1874: 39 [23]. Type species by subsequent designation (Villiers, 1948: 55 [22]): *Reduvius marginellus* Fabricius, 1803. As subgenus of *Reduvius*.**Note:** Hsiao and Ren in 1981 keyed ten species and two subspecies of *Rhynocoris* in China [3] but arranged all Chinese *Rhynocoris* species under the Neotropical generic name, *Harpactor* Laporte [3,19]. Most of the non-neotropical species in *Harpactor* belong in *Rhynocoris* and other genera, such as *Sphedanolestes* Stål, 1867 and *Biasticus* Stål, 1867 [1,2,3].


**The key to the Chinese species in the genus *Rhynocoris* Hahn, 1834**




1.Anterior pronotal lobe with dense appressed setae and glabrous arc-shaped sculptures ………………………………………………………………………………………..2
– Anterior pronotal lobe smooth, without dense appressed setae and glabrous sculptures…………………………………………………………………………………………3
2.Body relatively long and robust, over 14.5 mm in males and 16.5 mm in females in length; anterior pronotal lobe with deep glabrous sculptures………………………………………………………………………………………………………………………………………………*Rhynocoris incertis* (Distant, 1903)

–Body relatively short and slender, less than 14.0 mm in males and 16.0 mm in females in length; anterior pronotal lobe with shallow glabrous sculptures……………………………………………………………………………………………………………………………………….*Rhynocoris marginellus* (Fabricius, 1803)

3.Second antennal segment longer than third segment………………………………………………………………………………………………………………………………….4
– Second antennal segment shorter than third segment…………………………………………………………………………………………………………………………………..9
4.Abdomen ventrally red to orange with black transversal intersegmental stripes………………………………………………*Rhynocoris minutus* Liu, Zhao & Cai **sp. nov.**
– Abdomen ventrally black, or red to orange without black transversal intersegmental stripes……………………………………………………………………………………5
5.Connexivum unicolor, reddish…………………………………………………………………………………………………………*Rhynocoris rubromarginatus* (Jakovlev, 1893)
– Connexivum bicolor, basal half of each segment black, apical half red………………………………………………………………………………………………………………6
6.All femora with wide red annular markings……………………………………………………………………………………………………………………………………………7
– All femora with brownish annular markings, or completely black……………………………………………………………………………………………………………………8
7.Each femur with single red annular marking basally………………………………………………………………………………………..*Rhynocoris altaicus* Kiritshenko, 1926
– Each femur with two red annular markings, basally and medially………………………………………………………………………..*Rhynocoris dauricus* Kiritshenko, 1926
8.Pronotum completely black; all femora black, with reddish brown to dark brown annular marking basally and medially……………*Rhynocoris leucospilus* (Stål, 1859)
– Pronotum black, sometimes lateral margin of lateral pronotal angle red; all femora completely black…………………………………*Rhynocoris sibiricus* (Jakovlev, 1893)
9.Connexivum black; body entirely black, except red coxae and trochanters……………………………………………………… …………..*Rhynocoris reuteri* (Distant, 1879)
– Connexivum bicolor, red and black; body red with black markings………………………………………………………………………………………………………………..10
10.Femora red, with irregular black annular markings…………………………………………………………………………………………………………………………………11
– Femora black, without annular markings………………………………………………………………………………………………………………………………………………12
11.Body relatively long, more than 18 mm in length; ventral surface of abdomen completely black; second antennal segment slightly longer than or approximately equal to third segment………………………………………………………………………………………………………………………….*Rhynocoris monticola* (Oshanin, 1871)

–Body relatively short, less than 18 mm in length; ventral surface of abdomen red, with black longitudinal markings in middle part and on both sides; second antennal segment noticeably shorter than third segment………………………………………………………………………………………*Rhynocoris iracundus* (Poda, 1761)

12.Femora black, ventral surface with a series of white spots; sterna of abdomen red, with yellowish-white and black intersegmental stripes……………………………………………………………………………………………………………………………………………………..*Rhynocoris costalis* (Stål, 1867)
– Femora completely black; sterna of abdomen red, with black intersegmental stripes……………………………………………………..*Rhynocoris fuscipes* (Fabricius, 1787)



***Rhynocoris costalis* (Stål, 1867)**
(Figure 1, Figure 2 and Figure 3)Chinese common name: 山彩瑞猎蝽*Reduvius costalis* Stål, 1867: 285 [25]. Syntype (1♀): “Bengalia”, NHRS.*Reduvius (Reduvius) costalis*: Reuter, 1883: 293 [26], as a variety of R. fuscipes.*Harpactor costalis*: Distant, 1904: 334 [27].*Harpactor fuscipes*: Hsiao and Ren, 1981: 532 [3]. Misidentification.*Rhynocoris costalis*: Putshkov, V.G. and Putshkov, P.V., 1988: 165 [28]; Maldonado-Capriles, 1990: 278 [1]; Putshkov, V.G., Putshkov, P.V. 1996: 247 [2].**Type material examined. Syntype**: 1♀, Bengalia, Sundwall (NHRS) (Figure 1).

**Additional materials examined. China**: 1♂1♀, Guangdong, Zhanjiang, Lianjiang, Niushili, 21.8315° N, 110.0279° E, 74.7 m, 2022-VII-16–17, Chen Ting; 2♂, Hainan, Sansha, Yongxing, 16.8342° N, 112.3439° E, 2015-XI-16, Wu Qingtao; 1♂, Hainan, Dongfang, Datian, 19.1105° N, 108.7851° E, 62 m, 2009-VIII-2, Shi Aimin; 1♀, Guangxi, Nanning, Fusui, 2004-VIII-19, 200 m, Zhang Kuiyan; 1♂, Guangxi, Nanning, Wuming, Lijian, 2012-IX-27, Xu Huanli; 2♂2♀, Guangxi, Longlin, Jinzhongshan, 2014-VII-29; 1♀, Guangxi, Liuzhou, 2002-VIII-10, Liao Chaosu; 1♀, Yunnan, Jingdong, 2005-IV-30, Wang Hesheng; 1♀, Yunnan, Menglun, Lyushilin, 21.9088° N, 101.2814° E, 2009-V-5, Bai Xiaoshuan (CAU). **Vietnam**: 1♀, Hanoi, Tu Liem, 1996-III-26, Truong Xuan Lam; 2♂1♀, Hòa Bình, Đa Phúc-Yên Thủy, 1999-VIII-3, Truong Xuan Lam; 1♂, Hòa Bình, Đa Phúc-Yên Thủy, 1999-VIII-3, Truong Xuan Lam (CAU). **Cambodia**: 1♂, Siem Reap, Angkor Thom, 2005-VI-10, Var and Hagebaert (RBINS).

**Figure 1 insects-16-00823-f001:**
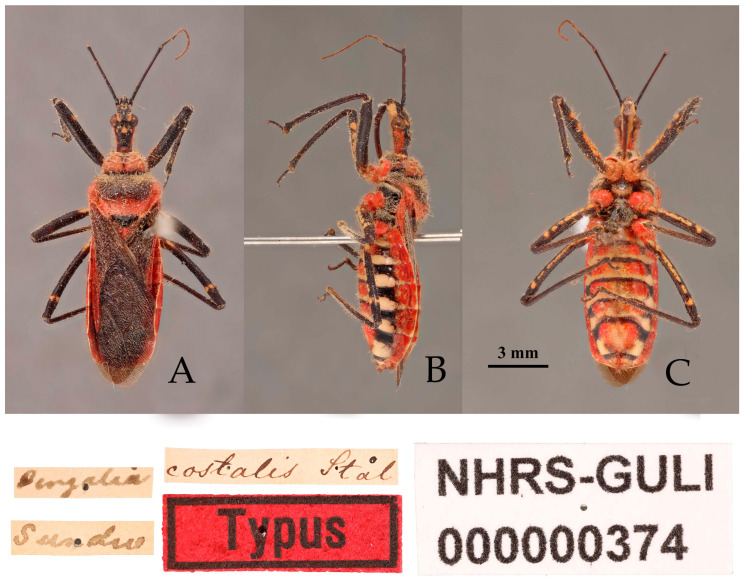
*Rhynocoris costalis* (Stål, 1867), female, syntype, habitus with labels. (**A**) dorsal view; (**B**) lateral view; (**C**) ventral view (©NHRS, photographed by Gunvi Lindberg).

**Redescription. Coloration.** Body black, with orange red to bright red and yellowish-white markings (Figure 1 and Figure 2).

Reddish-colored structures: Transversal stripe anterior to head constriction, postocular part, anterior lateral angle of pronotum, anterior pronotal lobe, most of posterior pronotal lobe (except transversal constriction and middle part of anterior area black), propleuron (except anterior angle and a large spot in middle part black), mesopleuron and metapleuron (except irregular markings of upper margin), thoracic sterna (except spots of middle part black), apical half of scutellum, most of corium (except inner sides black), each connexival segment (sometimes basal part and external margin black), coxal cavities, coxae, trochanters, ventral surface of abdomen (except intersegmental stripes and large spots of two lateral sides black and yellowish-white);

Yellowish-white-colored structures: Ventral surface of head, a small spot between ocelli, collar (except anterior lateral angles), spot of metapleuron, series of spots on ventral surface of femur, transversal stripe on posterior margin of each abdominal segment; 

Black-colored structures: Antennae, inner side of first antennal segment, second and third antennal segments, apical part of head, head vertex, two longitudinal stripes of ventral side of posterior lobe of head, transversal constriction of pronotum, large spot in middle part and anterior angle of propleuron, basal part of mesopleuron and metapleuron, femur (except yellowish-white spots), tibia, inner side of corium, clavus, distal part of trochanters, lateral sides of abdominal sterna (except white spots), transversal stripe of anterior margin of each abdominal sternum (Figure 1 and Figure 2).

**Figure 2 insects-16-00823-f002:**
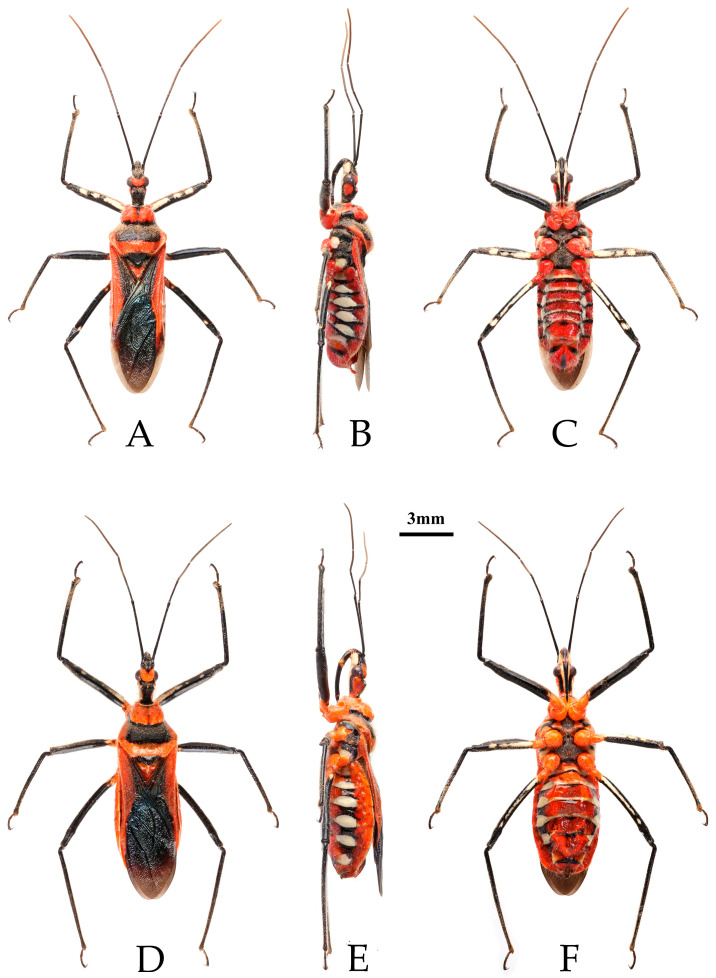
*Rhynocoris costalis* (Stål, 1867), non-type male and female, habitus. (**A**–**C**) male; (**D**–**F**) female. (**A**,**D**) dorsal view; (**B**,**E**) lateral view; (**C**,**F**) ventral view.

**Structure.** Body length 11.45–12.07 mm in male, 12.77–13.44 mm in female, elongated oval, covered with pale setae (Figure 1 and Figure 2). Posterior pronotal lobe, pleura and sterna of meso- and meta-thoraxes, corium and legs covered with pale, short, appressed setae. Head subequal to or slightly shorter than pronotum in length; anteocular portion subequal to or slightly shorter than postocular; second antennal segment shorter than third segment in male, slightly shorter than third in female; second labial segment about 1.5 times length of first. Anterior lateral angle of pronotum prominent, short and conical; anterior pronotal lobe with deep arc-shaped sculptures, median longitudinal sulcus deep, with two prominent round elevations on either side of posterior half; posterior pronotal lobe convex, central part slightly flattened; posterior angles of pronotum protruding backward; posterior margin of pronotum nearly straight. Fore wings distinctly surpassing tip of abdomen.

**Male genitalia.** Medial process of pygophore extending backward, apical part with a central concavity (Figure 3A,B); paramere rod-shaped, basal part curved, with a small protrusion on inner side of subapical part (Figure 3C–E). Phallobase of phallosoma short, basal plate bridge thicker than basal plate (Figure 3F); dorsal phallothecal sclerite with a strongly sclerotized margin around it, with a central concavity, an umbrella-shaped sclerotized support structure in middle part, and a transverse sclerotized plate in sub-basal part (Figure 3G–J); struts long, stout, and almost completely separated from each other (Figure 3H; endosoma with slender forked sclerite, and tips sharp (Figure 3G–J), apical part with two stripe-shaped sclerites on either side (Figure 3G–I).

**Measurements** [♂ (n = 11)/♀(n = 9), in mm]. Body length (from apex of head to tip of abdomen) 11.45–12.07/12.77–13.44; body length (from apex of head to tip of fore wing) 12.72–13.47/13.36–14.85. Length of head 2.76–2.87/2.80–3.01; length of anteocular region 0.97–1.03/0.92–1.04; length of postocular region 1.07–1.16/1.17–1.29; length of synthlipsis 1.28–1.35/1.32–1.37; interocellar space 0.71–0.75/0.70–0.79; length of antennal segments I–IV = 3.03–3.58/2.77–3.62, 1.13–1.37/1.34–1.75, 1.26–1.51/1.32–1.72, 2.98–3.75/3.27–3.94; length of visible labial segments I–III = 1.03–1.15/1.21–1.27, 1.61–1.81/1.86–1.93, 0.36–0.39/0.33–0.43. Length of anterior pronotal lobe 0.99–1.17/1.08–1.16; length of posterior pronotal lobe 1.65–1.74/1.70–1.95; length of pronotum 2.64–2.89/2.83–3.06; width of anterior pronotal lobe 1.78–1.95/1.87–2.07; width of posterior pronotal lobe 3.26–3.45/3.61–3.70; basal width of scutellum 1.48–1.70/1.71–1.82; length of scutellum 1.06–1.19/1.19–1.31; length of fore wing 8.47–8.75/9.34–9.83; length of fore femur, tibia, tarsus = 3.87–4.08/4.21–4.54, 4.40–4.68/4.98–5.11, 0.96–1.23/1.16–1.25; length of mid femur, tibia, tarsus = 3.03–3.50/3.36–3.56, 3.83–4.18/3.93–4.40, 0.89–1.11/0.92–1.10; length of hind femur, tibia, tarsus = 4.32–4.74/4.52–4.79, 5.60–6.02/5.97–6.44, 0.99–1.12/0.98–1.16; length of abdomen 5.63–5.81/6.30–6.64; maximum width of abdomen 3.01–3.79/3.11–4.21.

**Distribution.** China (Fujian, Guangdong, Guangxi, Hainan, Sichuan, Taiwan, Xizang, Yunnan) [3]; Bangladesh [25]; Cambodia (**new record**); India [3]; Indonesia (Java, Sumatra) [3]; Malaysia [3]; Myanmar [3]; Sri Lanka [3]; Vietnam (**new record**).

**Remarks.** This species *R. costalis* was first described by Stål [25] in the genus *Reduvius* Fabricius, 1775. Reuter [26] subsequently considered it as a variety of *Rhynocoris fuscipes*. These two species share similar morphological features, especially with almost no distinguishable differences when viewed dorsally, so their identification can sometimes cause confusion (e.g., Hsiao and Ren [3]). Putshkov and Putshkov [2] correctly recorded the distribution of both species in China. The two species are confused concerning their distribution and taxonomy in the Chinese literature [3]. However, given their significance as known natural enemies in agricultural pest control [3,5,6,7,12,13,14], it is essential to clarify their identities.

**Figure 3 insects-16-00823-f003:**
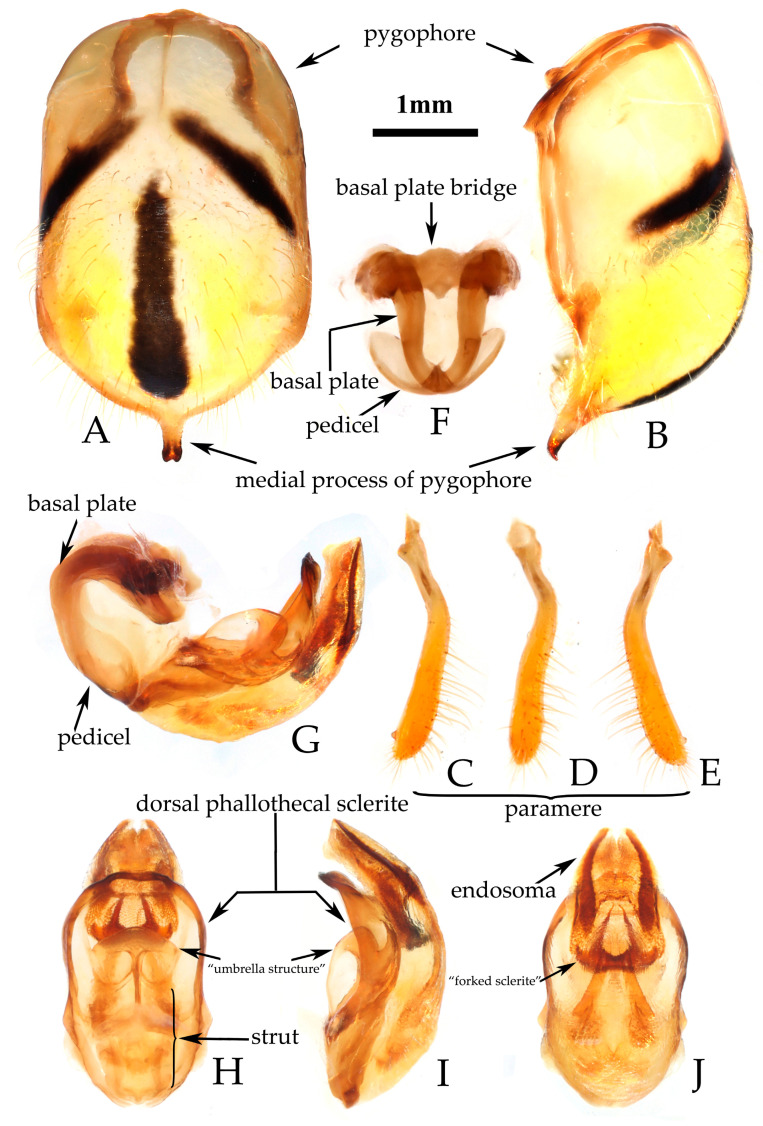
*Rhynocoris costalis* (Stål, 1867), male genitalia. (**A**,**B**) pygophore; (**C**–**E**) right paramere; (**F**) phallobase; (**G**) phallus (aedeagus); (**H**–**J**) phallosoma; (**A**,**J**) ventral view; (**B**,**G**,**I**) lateral view; (**H**) dorsal view.

One female specimen deposited in the collection of NHRS (NHRS-GULI000000374, Figure 1), matching the original description and bearing a red “Typus” label and Stål’s identification label, was examined based on high-quality photographs in the course of this study. It is recognized as a syntype of this species. The yellowish-white stripes on the ventral surfaces of the femora and abdomen are characteristic of the species, which are lacking in *R. fuscipes* (see below in Remark region of *R. fuscipes*). In most harpactorines, the dorsal phallothecal sclerite is generally a flat, nearly elliptical layer of sclerotized plate, and the apical part of the endosoma is armed with many spines. The structure of the male genitalia of *R. costalis* and *R. fuscipes* is complex and unique among members of the reduviid subfamily Harpactorinae, yet different among the two species. In *R. costalis*, the specialized structure of the dorsal phallothecal sclerite is “W”-shaped, but umbrella-shaped in *R. fuscipes*. We provided the detailed illustrations and descriptions of the male genitalia for these two species.

In *R. costalis*, some individuals have a few red or white spots on the labium, while others have a completely black labium. 


***Rhynocoris fuscipes* (Fabricius, 1787)**
(Figure 4, Figure 5 and Figure 6)Chinese common name: 红彩瑞猎蝽*Reduvius fuscipes* Fabricius, 1787: 312 [29]. Syntypes (2♀): “India Orientalis”, ZMUC.*Reduvius sanguinolentus* Wolff, 1804 [30]: 166. Syntype(s): India, depository unknown. Synonymized by Stål, 1859: 203 [31].*Reduvius corallinus* Lepeletier and Serville, 1825: 279 [32]. Syntype(s): India, depository unknown. Synonymized by Stål, 1874: 39 [23].*Reduvius (Reduvius) fuscipes*: Stål, 1874: 39 [23].*Harpactor fuscipes*: Walker (1873: 110 [33]).*Harpactor bicoloratus* Kirby, 1891: 120 [34]. Syntype (1♀): Sri Lanka, Southern Prov., Hambantota, NHMUK. Synonymized by Distant (1903: 205 [35]).*Rhinocoris fuscipes*: Bergroth, 1914: 362 [36].*Rhynocoris fuscipes*: Maldonado-Capriles, 1990: 279 [1]; Putshkov and Putshkov, 1996: 248 [2].**Type material examined. Syntypes**: 2♀, “India Orientalis” (ZMUC) (Figure 4).

**Additional materials examined. China**: 1♂, Yunnan, Yuxi, Xinping, Pingdian, Mopanshan, 24.0029° N, 101.9575° E, 1860 m, 2022-VIII-4, Zhu Pingzhou and Wang Xinkai; 1♀, Yunnan, Wenshan, Qiubei, Shupi, Xinan, 23.9136° N, 104.2269° E, 1538 m, 2022-VII-30, Zhu Pingzhou and Wang Xinkai; 3♂, Yunnan, Kunming, Panlong, Xiaohe, 2006-VII-29, Dong Baoxin; 3♂4♀, Yunnan, Nanjian, Huilong, 2005-V-5, Wang Hesheng; 1♂, Yunnan, Dali, Weishan, 2006-VIII-10, Wang Hesheng; 1♀, Yunnan, Baoshan, Tengchong, Mazhan, 2005-VIII-17, Li Tingjing; 1♂, Yunnan, Huaping, Haba, 2016-VII-30, Sun Ziqiang; 1♀, Guizhou, Weining, Ertang, 1979-VII-12 (CAU). **India**: 1♀, Andhra Pradesh, Nellore, Naidupet, Dwarakapuram, 2016-IX-1, Liu W.-T. (CAU). **Sri Lanka**: 1♀, Southern Prov., Hambantota, E.E. Green (NHMUK, syntype of *H. bicoloratus*).

**Redescription. Coloration.** Body generally bright red to deep red, with black stripes, somewhat shiny (Figure 4 and Figure 5).

Reddish-colored structures: Ventral surface of head, transverse stripe between eyes, round spot behind eyes, anterior angles of pronotum, anterior pronotal lobe (except median longitudinal sulcus), lateral and posterior parts of posterior pronotal lobe, apical half of scutellum, prosternum, coxal cavities, coxae, outer part of corium, abdomen (except black markings); 

Black-colored structures: Antennae, head vertex, two longitudinal stripes of ventral side of posterior lobe of head, collar (except anterior lateral angle red), median longitudinal sulcus and posterior margin of anterior pronotal lobe, anterior 2/3 of central part of posterior pronotal lobe, basal half of scutellum, mesosternum, metasternum, propleuron (except red markings and coxal cavities), mesopleuron, and metapleuron (except coxal cavities), legs (except coxae), inner part of corium, clavus, membrane, transversal intersegmental stripes of abdominal sterna, basal part of each connexival segment, median longitudinal stripe and two lateral oblique stripes of pygophore; 

Other coloration structure: Small round spot between ocellus light yellow to yellow (Figure 4 and Figure 5).

**Figure 4 insects-16-00823-f004:**
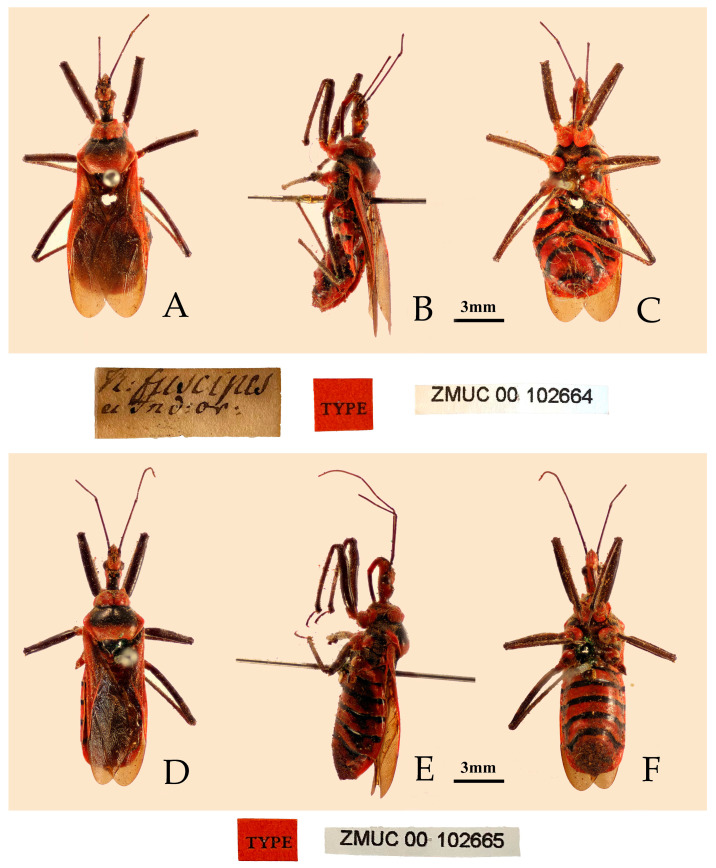
*Rhynocoris fuscipes* (Fabricius, 1787), female, two syntypes, habitus with labels. (**A**,**D**) dorsal view; (**B**,**E**) lateral view; (**C**,**F**) ventral view (©ZMUC, photographed by Sree G. Selvantharan and Lars Vilhelmsen).

**Structure.** Body medium-sized, elongated oval, somewhat robust (Figure 4 and Figure 5). Posterior pronotal lobe, pleura and sterna of meso- and meta-thoraxes, corium and legs covered with pale, short, appressed setae. Head subequal to pronotum in length; anteocular portion subequal to postocular in length; second antennal segment slightly longer than or subequal to third segment; first antennal segment subequal to fourth segment or combined length of second and third segments. Collar narrow, anterior lateral angles short and conical; anterior pronotal lobe with arc-shaped sculptures, posterior half of median longitudinal sulcus deep, with two prominent round elevations on either side; posterior pronotal lobe slightly rough in central region; posterior angles slightly protruding; posterior margin nearly straight; fore wing surpassing abdominal tip. Abdomen slightly laterally expanded.

**Figure 5 insects-16-00823-f005:**
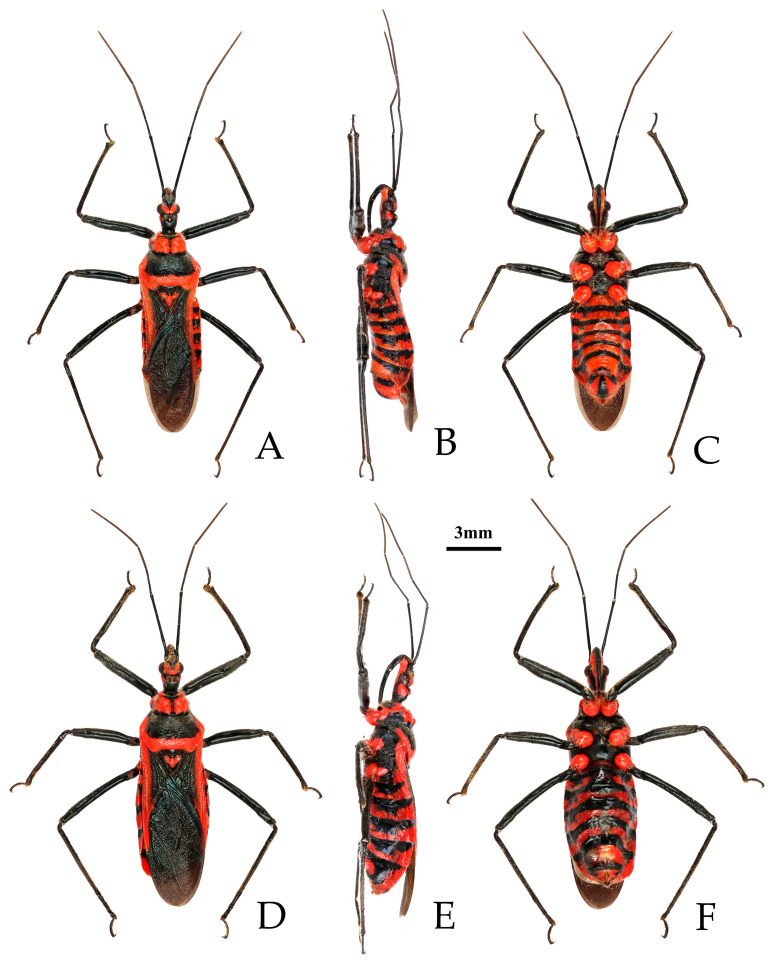
*Rhynocoris fuscipes* (Fabricius, 1787), non-type male and female, habitus. (**A**–**C**) male; (**D**–**F**) female; (**A**,**D**) dorsal view; (**B**,**E**) lateral view; (**C**,**F**) ventral view.

**Male genitalia**. Medial process of pygophore extending backward with a narrow tip, its apical part with a central concavity (Figure 6A,B); paramere rod-shaped, basal part curved, apical part blunt and rounded, with a noticeable protrusion on inner side of subapical part (Figure 6C,D). Phallobase of phallosoma stubby, basal plate bridge subequal to basal plate in length and thickness (Figure 6E); dorsal phallothecal sclerite with a strongly sclerotized margin around it, a "W"-shaped sclerotized support structure in middle part, and a transverse sclerotized plate in sub-basal part (Figure 3F–I); struts stout, completely separated from each other (Figure 3G); endosoma with stout forked sclerite, its tips blunt and rounded (Figure 3G), and apical part with two stripe-shaped sclerites on either side (Figure 6F–I).

**Figure 6 insects-16-00823-f006:**
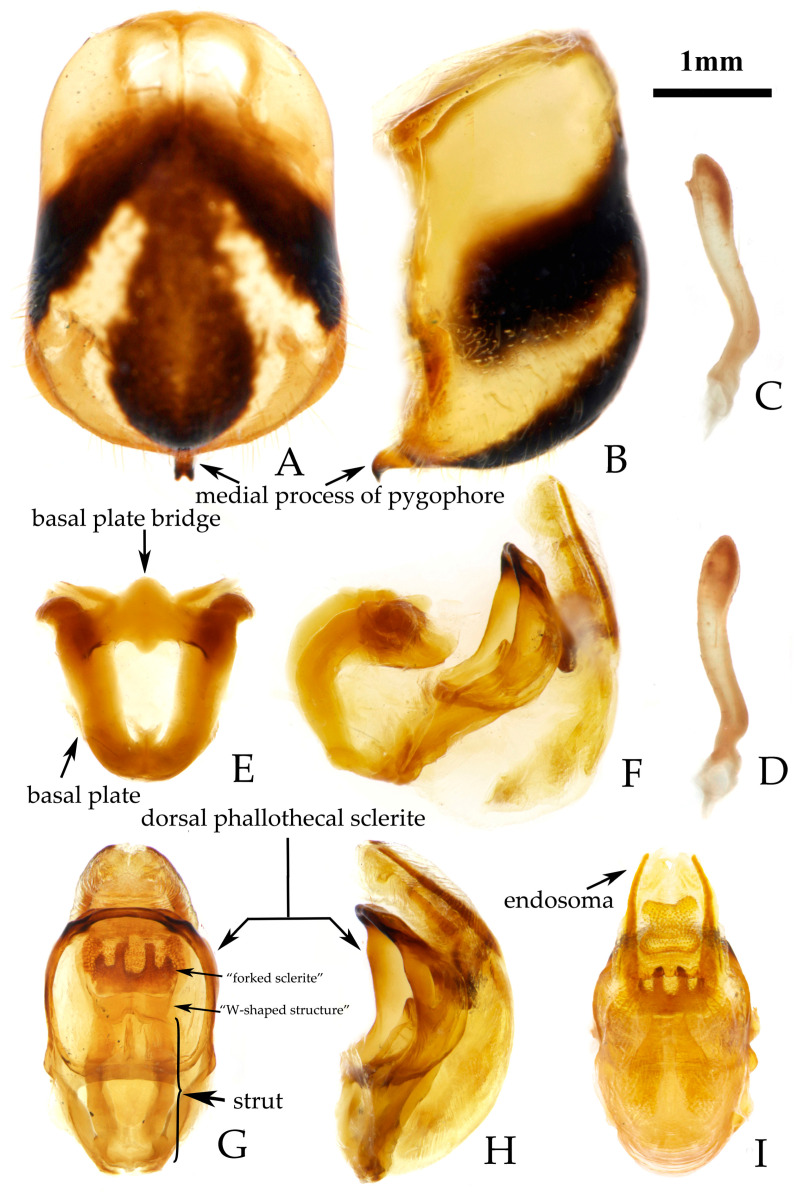
*Rhynocoris fuscipes* (Fabricius, 1787), male genitalia. (**A**,**B**) pygophore; (**C**,**D**) left paramere; (**E**) phallobase; (**F**) phallus (aedeagus); (**G**–**I**) phallosoma. (**A**,**I**) ventral view; (**B**,**F**,**H**) lateral view; (**G**) dorsal view.

**Measurements.** [♂ (n = 9)/♀ (n = 8), in mm]. Body length (from apex of head to tip of abdomen) 11.09–12.10/12.18–14.26; body length (from apex of head to tip of fore wing) 12.06–13.81/13.20–15.82. Length of head 2.41–2.83/2.59–2.95; length of anteocular region 0.82–0.94/0.96–1.06; length of postocular region 1.01–1.23/0.90–1.15; length of synthlipsis 1.27–1.41/1.31–1.41; interocellar space 0.65–0.76/0.65–0.73; length of antennal segments I–IV = 2.88–3.47/2.73–3.48, 1.28–1.58/1.31–1.51, 1.32–1.53/1.22–1.51, 2.56–3.45/2.38–3.26; length of visible labial segments I–III = 0.93–1.10/0.96–1.21, 1.59–1.75/1.70–2.05, 0.37–0.48/0.42–0.49. Length of anterior pronotal lobe 1.01–1.16/0.95–1.18; length of posterior pronotal lobe 1.48–1.93/1.63–1.85; length of pronotum 2.51–2.94/2.58–3.01; width of anterior pronotal lobe 1.86–1.97/1.91–2.02; width of posterior pronotal lobe 3.12–3.61/3.44–3.98; basal width of scutellum 1.60–1.84/1.72–1.94; length of scutellum 1.02–1.33/0.99–1.38; length of fore wing 7.76–9.27/8.56–10.52; length of fore femur, tibia, tarsus = 3.61–4.03/3.67–4.22, 4.04–4.71/4.32–5.41, 0.89–1.17/0.91–1.18; length of mid femur, tibia, tarsus = 3.02–3.46/3.05–3.98, 3.46–4.02/3.72–4.63, 0.79–1.02/0.83–1.05; length of hind femur, tibia, tarsus = 3.88–4.57/3.75–5.35, 5.05–6.12/5.01–6.72, 0.96–1.16/0.93–1.19. Length of abdomen 5.11–5.98/6.36–7.43; maximum width of abdomen 3.21–4.03/3.65–4.98.

**Distribution.** China (Fujian, Guizhou, Hainan, Yunnan) [11]; Cambodia [11]; Indonesia [31]; India [29]; Laos [11]; Myanmar [11]; Philippines [23]; Sri Lanka [31]; Thailand [11]; Vietnam [11].

**Remarks.** This species was described based on the specimens from “India Orientalis” [29] (Figure 4), and several synonymies were assigned to it subsequently [1,2]. Two female specimens deposited in the collection of ZMUC are recognized as syntypes of this species (ZMUC 00 102664 and 102665, Figure 4). Based on the original description [29] and the examination of the photographs of the syntypes, it could be confirmed that the records of *R. fuscipes* in China proposed by Hsiao and Ren [3] are the misidentification of *R. costalis*. The species *R. fuscipes* is most similar to *R. costalis* in terms of morphology, body size, and coloration. However, the two species can still be distinguished by the following characteristics: in *R. costalis*, body beneath has yellowish white spots on the ventral surface of the femora and the abdomen, while in *R. fuscipes*, body lacks yellowish-white spots, the femora are completely black, and the ventral surface of the abdomen only has red and black stripes without yellowish white series of spots. The structure of the male genitalia in both species is noticeably different. In *R. costalis*, the central part of the dorsal phallothecal sclerite has an umbrella-shaped sclerotized support structure, and the forked sclerites of the endosoma are slender and sharply pointed at the ends. In *R. fuscipes*, the central part of the dorsal phallothecal sclerite has a “W”-shaped sclerotized support structure, and the forked sclerites of the endosoma are somewhat shorter and thicker, with blunt, rounded ends. The two species also differ in the size of the protrusions on the inner side of the subterminal portion of the paramere, as well as in the direction in which the median process extends from the pygophore.

Most of the examined Chinese specimens of *R. fuscipes* have a uniformly black labium, while one specimen from India presents red at the base of its labium.


***Rhynocoris minutus* Liu, Zhao & Cai sp. nov.**
(Figure 7 and Figure 8)Chinese common name: 小瑞猎蝽**Type material. Holotype**: ♂, China, Yunnan, Lijiang, Yulong, Yulong Snow Mountain, 27.1129° N, 100.1552° E, 3121 m, 2020-VII-7, Chen Zhuo (CAU). **Paratypes**: 1♂, China, Yunnan, Diqing, Shangri-la, Sanba, Haba, 27.2255° N, 100.0616° E, 3176 m, 2020-VI-19, Huang Weidong, Guo Qiuhong, Liu Liyuan and Peng Feng; 1♀, China, Sichuan, Ganzi, Daocheng, Sangdui, 29.1410° N, 100.0852° E, 4282 m, 2016-VI-16, Lin Yejie; 1♀, Yunnan, Diqing, Gezan, 3000 m, 2010-VIII-18, Cao Liangming; 1♀, China, Yunnan, Lijiang, Yulong, Baisha, Wenhai, 3099 m, 2020-VI-16, 26.9666° N, 100.1706° E, Huang Weidong; 1♀, China, Yunnan, Lijiang, Yulong, 1980-VIII-29; 1♀, China, Yunnan, Lijiang, Yulong, Ludian, Kuaihuoyuan, 3157 m, 2020-VII-3, 27.1949° N, 99.4268° E, Chen Zhuo (CAU).

**Figure 7 insects-16-00823-f007:**
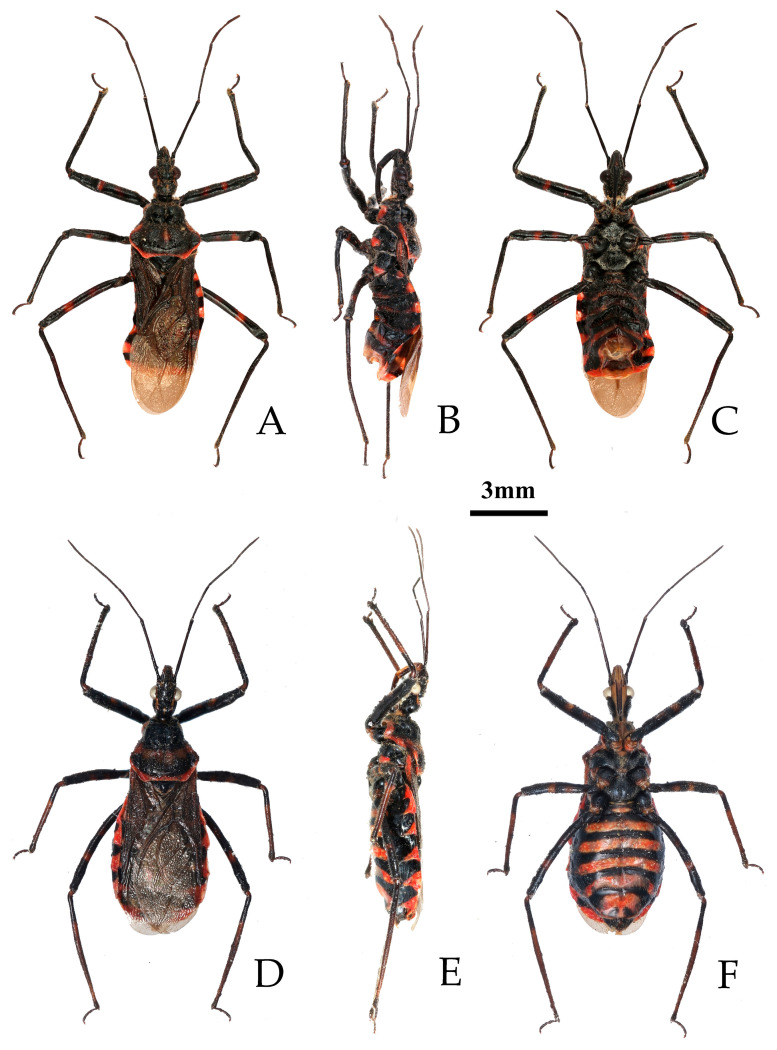
*Rhynocoris minutus* Liu, Zhao & Cai sp. nov., habitus. (**A**–**C**) male, holotype; (**D**–**F**) female, paratype; (**A**,**D**) dorsal view; (**B**,**E**) lateral view; (**C**,**F**) ventral view.

**Diagnosis**. The new species is similar to *Rhynocoris marginellus* and *R. incertis* in the general body shape and the genital structure, but in the latter two species, the body length is more than 11 mm in males and 14 mm in females, and the ventral abdominal surface has no transversal intersegmental stripes. The new species can be distinguished from the two species and its other Chinese *Rhynocoris* congeners by the following morphological characters: the body is significantly smaller, less than 10 mm in males and 12 mm in females, the ventral side of abdomen is red to orange with black regular transversal intersegmental stripes. The new species is only distributed in high-altitude areas of the Hengduan Mountains in Southwest China.

**Description. Coloration.** Body black, with red spots or stripes (Figure 7).

Reddish-colored structures. Posterior and lateral margins of posterior pronotal lobe, marginal stripe of fore coxal cavity, two obscure annular markings of middle part of femora, posterior half of each connexival segment, transversal stripe of anterior margin of second abdominal sternum, and transversal stripes of posterior halves of third to seventh abdominal sterna (except middle part of seventh segment black), basal part of anterior margin of fore wing; median longitudinal part of posterior pronotal lobe scattered with red;

Other coloration structures: Ventral side of head orange; spot between eyes and ipsilateral ocellus yellowish brown; apical part of scutellum yellow; first labial segment, most of first antennal segment (except basal and apical parts), second to fourth antennal segments, tarsi brown to dark reddish brown; tibiae (except basal and apical parts black) dark brown to black-brown (Figure 7).

**Structure.** Body small-sized, elongate-oval, robust. Body sparsely covered with short, erect setae (Figure 7); head, thorax, sides of second abdominal sternum densely covered with yellowish-white, short, appressed setae; legs sparsely covered with pubescence and setae of varying lengths. Postocular portion of head rounded and bulging, subequal to or slightly longer than anteocular portion; second antennal segment significantly longer than third segment, and first antennal segment slightly shorter than combined length of second and third antennal segments. Anterior lateral angles of pronotum prominent and short-conical; median longitudinal sulcus of anterior pronotal lobe prominent; posterior pronotal lobe medianly with a longitudinal sulcus; posterior lateral angles of pronotum rounded; posterior angles blunt and rounded; posterior margin slightly concave. Fore wing surpassing abdominal tip (Figure 7). Abdomen laterally roundly expanded (Figure 7).

**Male genitalia.** Medial process of pygophore slightly thick and short, apical part concave in middle part, with two short conical projections on both sides (Figure 8A,B). Paramere clavate, basal part distinctly curved, middle part slightly curved, and covered with erect short setae (Figure 8C–E). Basal plate somewhat slender and elongated, with a thin basal plate bridge, basal plate prolongation short (Figure 8F); dorsal phallothecal sclerite short and only covering basal 3/5 of phallosoma (Figure 8G,H); lateral arms of dorsal phallothecal sclerite strongly sclerotized (Figure 8G); lateral phallothecal sclerite with two darkly sclerotized longitudinal bands (Figure 8G–K); apical part of endosoma armed with a pair of groupings of small denticles (Figure 8I–K).

**Figure 8 insects-16-00823-f008:**
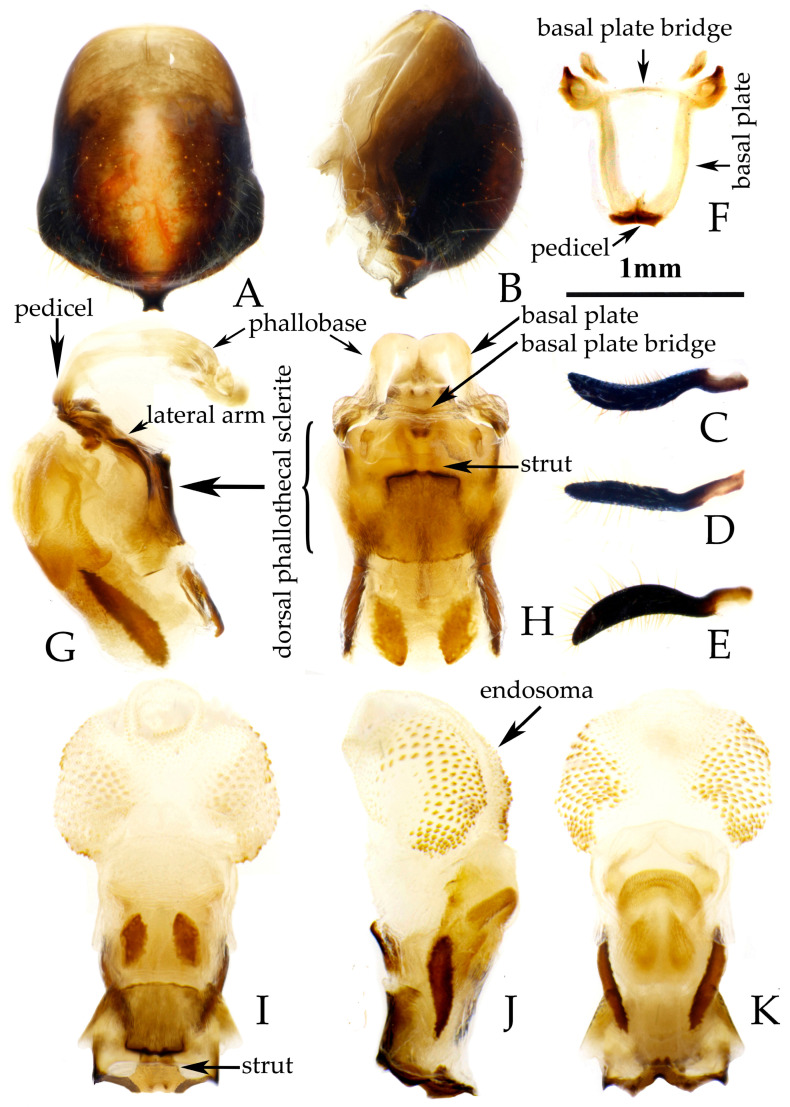
*Rhynocoris minutus* Liu, Zhao & Cai sp. nov., male genitalia. (**A**,**B**) pygophore; (**C**–**E**) right paramere; (**F**) phallobase; (**G**,**H**) phallus (aedeagus); (**I**–**K**) natural extending condition of phallosoma. (**A**,**K**) ventral view; (**B**,**G**,**J**) lateral view; (**H**,**I**) dorsal view.

**Measurements** [♂ (n = 2)/♀ (n = 5), in mm, holotype in parentheses]. Body length (from apex of head to tip of abdomen) 8.23–8.25/9.92–10.87 (8.25); body length (from apex of head to tips of fore wings) 9.42–9.62/10.40–11.39 (9.62). Length of head 1.83–2.02/2.09–2.35 (2.02); length of anteocular region 0.68–0.69/0.78–0.88 (0.69); length of postocular region 0.74–0.84/0.77–1.01 (0.84); length of synthlipsis 1.14–1.17/1.12–1.35 (1.14); interocellar space 0.55–0.57/0.55–0.66 (0.57); length of antennal segments I–IV = 2.22–2.26/2.21–2.48 (2.22), 1.21–1.23/1.22–1.41 (1.21), 0.85–1.02/1.06–1.26 (0.85), 1.43/1.38–1.71 (1.43); length of visible labial segments I–III = 0.78–0.83/0.92–1.06 (0.78), 1.19–1.26/1.28–1.46 (1.26), 0.27–0.32/0.35–0.47 (0.32). Length of anterior pronotal lobe 0.72–0.83/0.87–0.98 (0.83); length of posterior pronotal lobe 1.06–1.12/1.11–1.44 (1.06); length of pronotum 1.84–1.89/1.99–2.31 (1.89); width of anterior pronotal lobe 1.35–1.46/1.41–1.65 (1.46); width of posterior pronotal lobe 2.40–2.55/2.68–2.94 (2.55); basal width of scutellum 1.09–1.12/1.06–1.43 (1.09); median length of scutellum 0.64–0.67/0.58–0.84 (0.64); length of fore wing 6.11–6.16/6.23–7.46 (6.16); length of fore femur, tibia, tarsus = 2.78–2.83/2.88–3.32 (2.78), 3.31–3.43/3.06–3.61 (3.43), 0.66–0.72/0.59–0.69 (0.72); length of mid femur, tibia, tarsus = 2.21–2.41/2.42–2.69 (2.21), 2.82–2.94/2.89–3.26 (2.82), 0.69/0.63–0.67 (0.69); length of hind femur, tibia, tarsus = 3.24–3.25/3.49–3.74 (3.25), 4.02–4.35/4.54–4.95 (4.35), 0.70–0.74/0.71–0.81 (0.74). Length of abdomen 4.01–4.34/4.99–5.81 (4.34); maximum width of abdomen 2.31–2.59/3.59–4.04 (2.59).

**Etymology.** The specific epithet is derived from the Latin adjective *minutus* (meaning small or smaller), referring here to the relatively small body size of this species.

**Distribution.** China (Sichuan, Yunnan).

### 3.2. Biology

#### 3.2.1. *Rhynocoris costalis* (Stål, 1867)

We recorded the images of the complete life history of *Rhynocoris costalis* (Stål, 1867) based on more than 20 live specimens collected from Shaoguan, Guangdong, China. One of the adult females in the experiment laid eight eggs at one time and formed an egg mass. In the laboratory, they usually lay their eggs on the rough surfaces of egg cartons (Figure 9A). Different oviposition substrates have significant effects on the oviposition biological characteristics of this species, such as pre-oviposition period, oviposition duration, frequency of oviposition, the fecundity per female, and hatching rate [37]. There are five nymphal instars (Figure 9B–F). Adults occur almost throughout the year, whereas the peak period is from July to September (Figure 9G–I). In Nanxiong town, Guangdong province of China, *R. costalis* has two generations a year, with adults beginning to overwinter from mid-to-late November until mid-to-early March of the following year and starting to lay eggs in mid-to-late March; the first generation hatching in the early April, emerging as adults in the late May, and starting to reproduce in mid-to-late July; and the adults of the second generation being visible in late September [38].

**Egg.** Yellowish brown, about 2.00 mm in length, 0.70 mm in width, and elongate oblong, with a slight luster. Each egg is adhered to others by a small amount of gel, thus forming an egg mass. The eggs gradually change from yellowish-brown to dark brown, with a nearly smooth surface, operculum is white (Figure 9A).

**Figure 9 insects-16-00823-f009:**
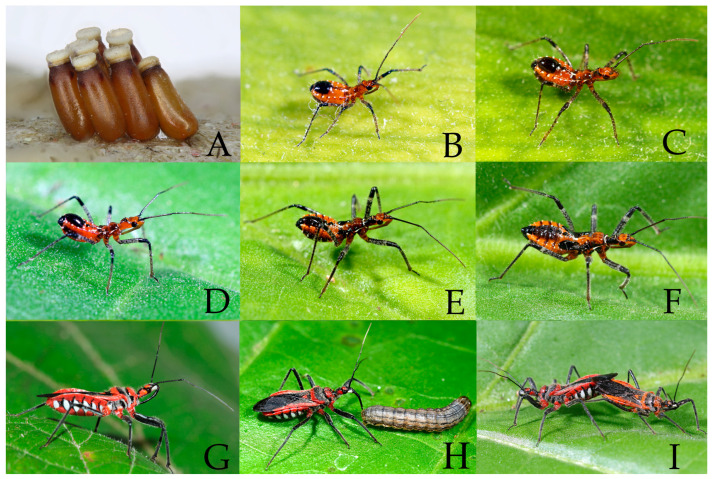
*Rhynocoris costalis* (Stål, 1867), (**A**) eggs; (**B**–**F**) nymphs; (**G**–**I**) adults. (**A**) egg mass; (**B**) first instar; (**C**) second instar; (**D**) third instar; (**E**) fourth instar; (**F**) fifth instar; (**G**) female; (**H**) female, feeding on larvae of the pest *Mythimna separata* in the laboratory; (**I**) female, left and male, right, mating tail to tail. (Photographed by Z.Y.C.).

**Coloration of nymph.** Orange red, with black and white markings (Figure 9B–F). Head is orange red, the black-spotted area in the postocular region gradually expands as the instar increases, and it spreads to the entire postocular region during the fourth to fifth instars. Most of the antennae are brown to brownish black, but the middle part of the first segment and the fourth segment are slightly lighter in color. Pronotum orange red to brownish red with black margins. Femora are yellowish white with black spots, subapical part is with a pale annular marking, and apical parts of femora are black. Tibiae are yellowish white with black spots, base of tibiae black. Tarsus is brown to black. Abdomen is orange red, with a large black spot on the posterior part of abdominal terga abdominal sterna are black with white spots on the ventral side. Wing pad is black.

**Structure of nymph.** Body sparsely is covered with pale setae. First and second antennal segments are sparsely covered with longer pale-yellow setae, while third and fourth antennal segments are densely with short, yellow setae. In first to third instar nymphs, the second antennal segment is significantly longer than the third, as the instar stage progresses, the third segment gradually lengthens, and by fifth instar nymphs, the second antennal segment becomes shorter than the third. Legs are sparsely covered with short, yellowish-white setae. Body slightly robust, posteriorly widened, each leg is slightly robust. The wing pad is visible in the third to fifth instar nymphs (Figure 9B–F).

**Life habit.** Newly hatched nymphs are gregarious and later disperse to move independently after the 2nd instar. Nymphs of all instars will molt after preying on their victims until their abdomens become swollen. Molting usually occurs on rough surfaces. Nymphs have the habit of cannibalism. The wing pads of the 5th instar nymphs swell strongly before eclosion. Adult insects have moderate flying ability.

**Feeding habit.** This species has a wide diet and can prey on a variety of farmland pests, such as *Spodoptera litura* (Fabricius, 1775) (Lepidoptera: Noctuidae) [38,39], *Agrotis ipsilon* (Walker, 1865) (Lepidoptera: Noctuidae) [39], *Myzus persicae* (Sulzer, 1776) (Hemiptera: Aphididae) [38,39], *Helicoverpa assulta* (Guenée, 1852) (Lepidoptera: Noctuidae) [38,39], *Spodoptera frugiperda* (Smith, 1797) (Lepidoptera: Noctuidae) [40].

#### 3.2.2. *Rhynocoris fuscipes* (Fabricius, 1787)

We recorded the life history of *Rhynocoris fuscipes* (Fabricius, 1787) based on about 20 live specimens collected from Zhaotong, Yunnan, China. One of the adult females in the experiment lays more than 20 eggs at one time and formed an egg mass (Figure 10A). In the laboratory, they usually lay their eggs on the rough surfaces of egg cartons (Figure 10A). There are five instars of nymphs (Figure 10B–F). In Virudhunagar District, Tamil Nadu, India, *Rhynocoris fuscipes* adults are observable across all months of the year, with the peak period occurring from April to July [41].

**Egg.** The egg is yellowish brown, about 2.00 mm in length, 0.70 mm in width, and elongate oblong, with an obvious luster. Each egg has an obvious layer of gel on its surface, which makes them stick to one another and thus form an egg mass. The eggs gradually change from yellowish-brown to dark brown, with a slightly rough surface, operculum is white to yellowish white.

**Coloration of nymph.** The nymphs are orange in *R. fuscipes* and orange red in *R. costalis*, but the color markings are very similar for the nymphs of two species. Body is orange, with black and white markings (Figure 10B–F). Head is orange, the black spot area of the postocular region gradually expands as the instar increases, and it spreads to the entire postocular region during the fourth to fifth instars. Eyes are black. In the first to third instar nymphs, the first antennal segment is orange (except the basal and apical parts), and it gradually turns black after the fourth to fifth instars. The second and third antennal segments black. Forth antennal segment orange to light brown. Thorax is orange with black markings. Femora and tibiae are yellowish white with black spots interspersed in the middle, and apical part of femora and basal part of tibiae are black. Tarsus is brown to black. Abdomen is orange, with white spots scattered on both the ventral and dorsal sides, and black markings on the posterior part of the dorsal side and on both sides of the ventral side. Wing pad is black. In live adults, the lateral sides of abdominal sterna are also with narrow white transversal stripes (Figure 10G), whereas in dried specimens, these white markings are nearly invisible (Figure 4 and Figure 5).

**Figure 10 insects-16-00823-f010:**
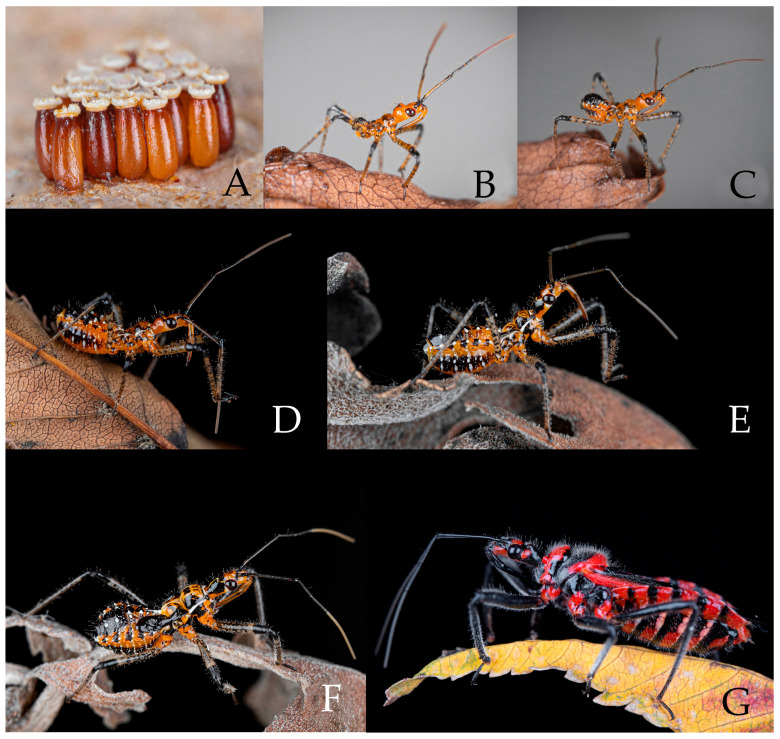
*Rhynocoris fuscipes* (Fabricius, 1787), (**A**) eggs; (**B**–**F**) nymphs; (**G**) adult. (**A**) egg mass; (**B**) first instar; (**C**) second instar; (**D**) third instar; (**E**) fourth instar; (**F**) fifth instar; (**G**) female. (Photographed by H.Y.X.).

**Structure of nymph.** Body is sparsely covered with pale setae. All segments of the antenna are densely covered with short, pale setae, while the first and second antennal segments are sparsely covered with long, brownish-yellow setae. In first to third instars of nymphs, the second antennal segment is significantly longer than the third, as the instar stage progresses, the third segment gradually lengthens, and by fifth instar nymphs, the length of the second antennal segment is not significantly longer than the third. Legs are sparsely covered with short, yellowish-white setae. Body is slightly robust, posteriorly widened, each leg is slightly robust. Third to fifth instar nymph are with visible wing pads (Figure 10B–F).

**Life habit.** *Rhynocoris fuscipes* has similar living habits to *Rhynocoris costalis*.

**Feeding habit.** This species has a relatively wide diet and can prey on approximately 42 species of agricultural pests, including *Spodoptera frugiperda* (Smith, 1797) (Lepidoptera: Noctuidae) [14], *Earias fabia* (Stoll, 1782) (Lepidoptera: Nolidae) [42], *Helicoverpa armigera* (Hübner, 1808) (Lepidoptera: Noctuidae) [42], *Dysdercus cingulatus* (Fabricius, 1775) (Hemiptera: Pyrrhocoridae) [14,43], *Spodoptera litura* (Fabricius, 1775) (Lepidoptera: Noctuidae) [14,44], *Nilaparvata lugens* (Stål, 1854) (Hemiptera: Delphacidae) [14,45], *Epilachna vigintioctopunctata* (Fabricius, 1775) (Coleoptera: Coccinellidae) [46], and so on.

## 4. Discussion

### 4.1. Taxonomy Comments and the Identification of Species

The two morphologically similar species, *R. costalis* and *R. fuscipes*, are widely distributed in the Oriental Region, and are common predatory natural enemy insects in the forest and agricultural ecosystem of East, Southern, and Southeast Asia [1,2,3,12,13,14,27,28,37,38,39,40,42,43,44,45,46]. The similarity between these two species in terms of body size, coloration, and morphology has long caused taxonomists confusion in their identification [3,26]. In the last Chinese literature about the two species [1,2,3,8,9,10,11,36], only the species name, *R. fuscipes,* is recorded. “A Handbook for the Determination of the Chinese Hemiptera-Heteroptera (II)” [3] especially has had a profound influence on the taxonomic study on the sub-order Heteroptera from China. In the book, Hsiao et al., in 1981, provided the description of the species *R. costalis*, but the given Latin name was *R. fuscipes*. From then until today, all research on these two species in China has been attributed to the Latin name of *R. fuscipes* [12,13,14,37,38,39,40]. The present study was conducted after examining the type specimens of the two species, *R. costalis* and *R. fuscipes,* to re-describe them and determine their taxonomic status, so as to facilitate people′s correct identification, and clarify the existence of their valid species, rather than a variant or synonym of *R. fuscipes* [11,36], and confirm that they both exist in China [2]. For over two hundred years, the distribution records for both widespread species have been inconsistent and unverified, leading to accurate and inaccurate reports in the literature. The present study provides accurate distribution data based on carefully examined and identified species.

Based on the consistent characteristic of white spots on the legs and abdomen, we have clarified the stable morphological features that distinguish these two species, and this distinction is also reflected in their scientific names: the Latin *costalis* means “rib-like or ribbed”, indicating the presence of white markings on the intersegmental areas of both sides of the abdomen beneath; *fuscipes* means “black legs”, referring to the entirely black legs of this species. On this basis, we have corrected the corresponding relationship between their scientific names and Chinese common names, which will have a positive impact on research related to biology, ecology, and pest control.

The new species *R. minutus* is distributed in the high-altitude regions of the Hengduan Mountains in southwestern China. In comparison with other *Rhynocoris* species native to China, this species has a significantly smaller body size, which could be associated with high-altitude conditions [47,48].

### 4.2. Effective Reduviid Predators in the Agricultural Ecosystem

Biological studies have shown that *R. costalis* and *R. fuscipes* are quite similar in the morphology of their eggs and nymphs, as well as in their biological habits and feeding habits. Their extensive distribution, large population, strong ecological adaptability [49], and broad feeding habits serve as the objective foundation for them to be widely used as natural enemy insects in biological control [38,39,40,41,42,43,44,45,46,49,50,51].

According to previous studies, they can prey on a variety of pests that pose a significant threat to agricultural production, such as *Spodoptera litura* (Fabricius, 1775) (Lepidoptera: Noctuidae) [38,39], *Spodoptera frugiperda* (Smith, 1797) (Lepidoptera: Noctuidae) [40], *Agrotis ipsilon* (Walker, 1865) (Lepidoptera: Noctuidae) [39], *Myzus persicae* (Sulzer, 1776) (Hemiptera: Aphididae) [38,39], *Helicoverpa armigera* (Hübner, 1808) (Lepidoptera: Noctuidae) [42], *Epilachna vigintioctopunctata* (Fabricius, 1775) (Coleoptera: Coccinellidae) [46], and *Nilaparvata lugens* (Stål, 1854) (Hemiptera: Delphacidae) [14,45]. This means that they can be released in fields of various crops, such as *Gossypium hirsutum* L. (cotton), *Solanum melongena* L. (brinjal), *Oryza sativa* L. (rice), *Arachis hypogaea* L. (groundnut), and *Nicotiana tabacum* L. (tobacco) [38,43,46,50,51].

Since our observations on the morphology and biology of *R. costalis* and *R. fuscipes* are largely based on specimens native to China, the morphological polymorphism and complex biological characteristics of these insects have not been fully elaborated. Their biological characteristics necessitate further observation to reveal more ecological and behavioral features. As important natural enemies commonly found in both natural ecosystems and agricultural ecosystems, there are still numerous aspects to be further studied in the future regarding the target pest species they regulate, their release techniques, and the mass rearing and propagation of these two species.

## Data Availability

The original contributions presented in this study are included in the article. Further inquiries can be directed to the corresponding authors.

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
