# Peer review of "Comments on Two Controversial Oriental Assassin Bug Species of the Genus Rhynocoris (Heteroptera: Reduviidae: Harpactorinae), with the Description of R. minutus sp. nov. from Chinaâ€"

_insects, 2025, doi:10.3390/insects16080823_

Round 1
Reviewer 1 Report
Comments and Suggestions for Authors
I have made several comments and suggestions along the manuscript.
An important issue: the necessity that the authors must point the structures of the phallus, to allow their recognition by the reader.
They stated the male genitalia of the studied species as "complex" and/or "unique". They must make clear why they considered them as such. Preferably by comparing with other species of other genera to justify their interpretation of the male genitalia.

I am not English native; therefore I am not really qualified to a complete answer to this item. However, in some portions of the description the authors used the verb "to be" excessively (e.g. "antennae are brown"). Descriptions must be concise and in these cases the verb does not need to be used, e.g.: - antenna brown - as simple as that will be better...
Author Response
For research article
|
Response to Reviewer 1 Comments
|
||
|
1. Summary |
|
|
|
Thank you very much for taking the time to review this manuscript. Please find the detailed responses below and the corresponding revisions in track changes in the re-submitted files. |
||
|
2. Questions for General Evaluation |
Reviewer’s Evaluation |
Response and Revisions |
|
Does the introduction provide sufficient background and include all relevant references? |
Yes |
Thank you |
|
Is the research design appropriate? |
Yes |
Thank you |
|
Are the methods adequately described? |
Yes |
Thank you |
|
Are the results clearly presented? |
Must be improved |
Thank you |
|
Are the conclusions supported by the results? |
Yes |
Thank you |
|
Are all figures and tables clear and well-presented? |
Must be improved |
Thank you |
|
3. Point-by-point response to Comments and Suggestions for Authors |
||
|
Comments 1: An important issue: the necessity that the authors must point the structures of the phallus, to allow their recognition by the reader.
|
||
|
Response 1: Thank you for pointing this out. I/We agree with this comment. Therefore, we have revised all the figures of the structures of the phallus. Page 8, Fig. 3: revised Page 12, Fig. 6: revised Page 16, Fig. 8: revised
|
||
|
Comments 2: They stated the male genitalia of the studied species as "complex" and/or "unique". They must make clear why they considered them as such. Preferably by comparing with other species of other genera to justify their interpretation of the male genitalia-Page 9, Paragraph 1, and line 259-260
“species is also different, and complex and unique in the members of the reduviid subfamily Harpactorinae. We provided the detailed...”
|
||
|
Response 2: Thank you for pointing this out. We agree with this comment. Therefore, we have compared the two species with other harpactorines-Page 9, Paragraph 1, and line 259-260
“species is also different, and complex and unique with the specialized structure of dorsal phallothecal sclerite, and “W”-shaped or umbrella-shaped structure of endosoma among the members of the reduviid subfa'mily Harpactorinae (vs. In most of harpactorines, the dorsal phallothecal sclerite generally is a flat, nearly elliptical layer of sclerotized plate, and the apicial part of endosoma is with many spines). We provided the detailed“
|
||
|
Comments 3:”occurring” -Page 1, Paragraph 1, and line 39
“in the Eastern Hemisphere, with only two occur in North America [1]. Most of them”
|
||
|
Response 3: Thank you for pointing this wrong word out. -Page 1, Paragraph 1, and line 39.
“Hemisphere, with only two species occurring in North America [1]. The majority of“
|
||
|
Comments 4: ”reaches”Page 2, Paragraph 1, and line 45, 46
“lobe, which reaching neither the collar anteriorly nor the transverse constriction 45 posteriorly [3, 4]. Adults of Rhynocoris are diurnal and can be found in varies vegetated“
|
||
|
Response 4: Thank you for pointing these two wrong words out. -Page 2, Paragraph 1, and line 45, 46 “reaches neither the collar anteriorly nor the transverse constriction posteriorly [3, 4]. Adults of Rhynocoris are diurnal and can be found in various vegetated habitats, feeding”
|
||
|
Comments 5: ” nov.”-Page 2, Paragraph 2, and line 60.
“we discovered a new species, R. minutus sp. nov, which has similar color pattern but smaller in size. The key is provided for the identification of the Rhynocoris species 61 occurring in China.”
|
||
|
Response 5: Thank you for pointing these two words out. -Page 2, Paragraph 2, and line 60
“Additionally, we discovered a new species, R. minutus sp. nov., which has similar color pattern but is smaller in size.”
|
||
|
Comments 6: Why approximately? Why couldn´t you proceed it in a exact time? “After examination,” -Page 2, Paragraph 8, and line 73-76
“Male genitalia were soaked in hot 10% NaOH solution for approximately five minutes to remove soft tissue, rinsed in distilled water, and dissected under a Motic binocular dissecting microscope. Dissected genitalia were placed in a vial with glycerin and, after examination, pinned under the corresponding specimen.”
|
||
|
Response 6: Thank you for pointing this out. The boiling time in hot 10% NaOH solution varies depending on the state of the insects, such as their freshness and size. Large and fresh specimens sometimes take 15 to 20 minutes to boiling while observing. Sometimes it takes several weeks of slow soaking to reach the best effect. Small and dry specimen maybe need to be examined every 5 minutes, just in case it's over-boiled.The main purpose is to remove the muscle and fat tissues, making the sclerites of the insect's genital clearly visible.-Page 2, Paragraph 8, and line 73-76
“Male genitalia were soaked in hot 10% NaOH solution for approximately five minutes to remove soft tissue, rinsed in distilled water, and dissected under a Motic binocular dissecting microscope. After examination, each dissected genitalia were placed in a vial with glycerin and pinned under the corresponding specimen.”
|
||
|
Comments 7: Include authorship of this species and its taxonomic position (Coleoptera: Family...)-Page 1, Paragraph 1 and line 89
“...mealworm, Tenebrio molitor. The eggs and 1–5 instars nymphs were described and...”
|
||
|
Response 7: Thank you for pointing this out. Page 3, Paragraph 1, and line 89
“...mealworm, Tenebrio molitor Linnaeus, 1758 (Coleoptera: Tenebrionidae). The eggs and 1–5 instars nymphs were described, and...”
|
||
|
Comments 8: Year of this reference...; Please, include authorship of these genera.-Page 3, Paragraph 8, and line 107, 110
“Note: Hsiao and Ren keyed 10 sepceis and 2 subspecies of Rhynocoris in China [3], but arranged all Chinese Rhynocoris species under the Neotropical generic name, Harpactor Laporte [3, 19]. Most of the non-neotropical species in Harpactor belong in Rhynocoris and other genera, such as Sphedanolestes, Biasticus [1–3].”
|
||
|
Response 8: Thank you for pointing these out.
“Note: Hsiao and Ren in 1981 keyed 10 species is and 2 subspecies of Rhynocoris in China [3], but arranged all Chinese Rhynocoris species under the Neotropical generic name, Harpactor Laporte [3, 19]. Most of the non-neotropical species in Harpactor belong in Rhynocoris and other genera, such as Sphedanolestes Stål, 1867 and Biasticus Stål, 1867 [1–3].”
|
||
|
Comments 9: All comments about the key are as followed -Pages 3-4, and lines 111-152
in total length [right?] Please check if you should better use "pedicel" here in order to follow the literature you cited in "mat. & met." as source of terminology. measuring first flagellomere? or "reddish"? Please check. Do you mean dark? Obscure is not very clear to define the coloration here. of all femora? of all femora? Please, make it clear. How long is it? The future reader must have a comparison to follow... all femora or only one of them (fore, middle, hind)? Please make it clear... Please, make it clear. How long is it? The future reader must have a comparison to follow... all femora, right? Make it clear and above, as commented there. Integument of... how long?
|
||
|
Respons 9: Thank you for pointing these out. All comments about the key have been revised.-Pages 3-4, and line 111-152: We think your opinion is particularly reasonable. The Heteropteran antenna consists of the pedicel (first antennal segment), pedicel (second antennal segment), and basiflagellum and distiflagellum (third and fourth antennal segments). However, in general, we only use first to fourth antennal segments in the description of morphological characters, which is even concise and easy to understand. We removed most of the size characters from the key. The revised key is following: “ The key to the Chinese species in the genus Rhynocoris Hahn, 1834 1. Anterior pronotal lobe with dense appressed setae and glabrous arc-shaped sculptures...2 -. Anterior pronotal lobe smooth, without dense appressed setae and glabrous sculptures...3 2. Body relatively long and robust, over 14.5 mm in male (14.8–15.5 mm) and 16.5 mm in female (17.0–17.8 mm) in length; anterior pronotal lobe with deep glabrous sculptures...Rhynocoris incertis (Distant, 1903) -. Body relatively short and slender, less than 14.0 mm in male (11.0–13.9 mm) and 16.0 mm in female (14.7–16.0 mm) in length; anterior pronotal lobe with shallow glabrous sculptures...Rhynocoris marginellus (Fabricius, 1803) 3. Second antennal segment longer than third segment...4 -. Second antennal segment shorter than third segment...9 4. Abdomen ventrally red to orange with black transversal intersegmental stripes...Rhynocoris minutus sp. nov. -. Abdomen ventrally black, or red to orange with irregular black markings ...5 5. Connexivum unicolor, reddish...Rhynocoris rubromarginatus (Jakovlev, 1893) -. Connexivum bicolor, basal half of each segment black, apical half red...6 6. All femora with wide red annular markings...7 -. All femora with vague annular markings or completely black...8 7. Each femur with single red annular marking basally...Rhynocoris altaicus Kiritshenko, 1926 -. Each femur with two red annular markings, basally and medially..Rhynocoris dauricus Kiritshenko, 1926 8. Pronotum completely black; all femora black, with pale annular marking basally and medially...Rhynocoris leucospilus (Stål, 1859) -. Pronotum black, sometimes lateral margin of lateral pronotal angle red; all femora completely black...Rhynocoris sibiricus (Jakovlev, 1893) 9. Connexivum black; body entirely black, except red coxae and trochanters...Rhynocoris reuteri (Distant, 1879) -. Connexivum bicolor, red and black; body red with black markings...10 10. Femora red, with irregular black annular markings...11 -. Femora black, without annular markings...12 11. Body relatively long, more than 18 mm in length; ventral surface of abdomen completely black; second antennal segment slightly longer than or approximately equal to third segment...Rhynocoris monticola (Oshanin, 1871) -. Body relatively short, less than 18 mm in length; ventral surface of abdomen red, with black longitudinal markings in middle part and on both sides; second antennal segment noticeably shorter than third segment...Rhynocoris iracundus (Poda, 1761) 12. Femora black, ventral surface with a series of white spots; sterna of abdomen red, with yellowish-white and black intersegmental stripes...Rhynocoris costalis (Stål, 1867) -. Femora completely black; sterna of abdomen red, with black intersegmental stripes...Rhynocoris fuscipes (Fabricius, 1787) ” |
||
|
Comments 10: you list too many parts or portions and only at the end their color. What about starting sentences with the color first, cited as:? It would be much better to the reader to read the color before reading all the portions with it first... it is really tiring reading coloration like this...
Pages 5,6, and lines 180-197; Pages 9, and lines 289-301; Pages 15, and lines 398-409.
|
||
|
Response 10: Thank you for pointing these out. We start sentences with the color first. Pages 5,6, and lines 180-197; Pages 9, and lines 289-301; Pages 15, and lines 398-409.
|
||
|
Comments 11: what does that mean to a future reader? You must define it more precisely-Page 6, Paragraph 2, and line 201 “Structure. Body medium-sized, elongated oval, covered with pale setae (Figures 1,...”
|
||
|
Response 11: Thank you for pointing this out.
“Structure. Body length 11.45–12.07 mm in male, 12.77–13.44 mm in female, medium-sized, elongated oval, covered with pale setae (Figures 1, 2)-Page 6, Paragraph 2, and line 201.”
|
||
|
Comments 12: What about to point these structures in the figures? It would allow the reader to understand exactly what you are describing... It is confusing without pointing them there...-Page 7, Paragraph 2, and lines 214-219
Please, point these structures in the figures! It would allow the reader to understand exactly what you are describing... It is confusing without pointing them there...-Page 11, Paragraph 3, and lines 325-328
If you do not point these structures in the figures, the reader will not be able to be sure about exactly which one they are...-Page 13, Paragraph 3, and lines 365-371
Point these structures in the figures, please.-Page 15, Paragraph 3, and lines 425-430.
|
||
|
Response 12: Thank you for pointing these out. We pointed these structures in the figures and marked the structural annotations of the male genital for Figures 3, 6, 8.
Page 7, Paragraph 2, and lines 214-219: revised. Page 11, Paragraph 3, and lines 325-328: revised. Page 13, Paragraph 3, and lines 365-371: revised. Page 15, Paragraph 3, and lines 425-430: revised.
|
||
|
Comments 13: Why? What characteristics made them so different? Complex and unique? Please, provide some other species in comparison...Page 8, Paragraph 2, and lines 259
“species is also different, and complex and unique in the members of the reduviid”
|
||
|
Response 13: Thank you for pointing this out. Page 8, Paragraph 2, and lines 259: revised.
“The structure of the male genitalia between these two species is also different, and complex and unique with the specialized structure of dorsal phallothecal sclerite, and “W”-shaped or umbrella-shaped structure of endosoma among the members of the reduviid subfamily Harpactorinae (vs. In most of harpactorines, the dorsal phallothecal sclerite generally is a flat, nearly elliptical layer of sclerotized plate, and the apicial part of endosoma is armed with many spines). “
|
||
|
Comments 14: Include authorship and taxonomic position (order: family) of all these species- Page 18, Paragraph 4, and lines 506, 507 Page 20, Paragraph 2, and lines 548-550 Page 20, Paragraph 4, and lines 581, 582 Page 21, Paragraph 1 and lines 587, 588
|
||
|
Response 14: Thank you for pointing this out. Page 18, Paragraph 4, and lines 506, 507: revised. Page 20, Paragraph 2, and lines 548-550: revised. Page 20, Paragraph 4, and lines 581, 582: revised. Page 21, Paragraph 1 and lines 587, 588: revised.
|
||
|
Comments 15: Comments about labium and rostrum. Rhynocoris As the first word of a sentence write the name of the genus in full, not abbreviated. labium [as the used in current terminology -Page 20, Paragraph 4, and lines 559-562
|
||
|
Response 15: Thank you for pointing this out. Agree.Throughout the MS, we have uniformly used labium or labial instead of rostrum or rostral. Page 20, Paragraph 4, and lines 559-562: revised.
|
||
|
4. Response to Comments on the Quality of English Language |
||
|
Point 1: The English could be improved to more clearly express the research. I am not English native; therefore I am not really qualified to a complete answer to this item. However, in some portions of the description the authors used the verb "to be" excessively (e.g. "antennae are brown"). Descriptions must be concise and in these cases the verb does not need to be used, e.g.: - antenna brown - as simple as that will be better...
|
||
|
Response 1: We try to revise.
|
||
|
5. Additional clarifications |
||
|
|
||

Reviewer 2 Report
Comments and Suggestions for Authors
Good manuscript, just few minor points to polish.

The English require a final reading by a native speaker.
Author Response
For research article
|
Response to Reviewer 2 Comments
|
||
|
1. Summary |
|
|
|
Thank you very much for taking the time to review this manuscript. Please find the detailed responses below and the corresponding revisions in track changes in the re-submitted files. |
||
|
2. Questions for General Evaluation |
Reviewer’s Evaluation |
Response and Revisions |
|
Does the introduction provide sufficient background and include all relevant references? |
Yes |
Thank you |
|
Is the research design appropriate? |
Yes |
Thank you |
|
Are the methods adequately described? |
Yes |
Thank you |
|
Are the results clearly presented? |
Yes |
Thank you |
|
Are the conclusions supported by the results? |
Yes |
Thank you |
|
Are all figures and tables clear and well-presented? |
Yes |
Thank you |
|
3. Point-by-point response to Comments and Suggestions for Authors |
|
|
|
Comments 1: Page 1, Paragraph 1, and lines 2,3: About the title: “Comments on two controversial oriental predatory species of the genus Rhynocoris (Hemiptera: Reduviidae: Harpactorinae), with the description of R. minutus sp. nov. from China” (1) “predatory”: This is general for all reduviids, lets delete here. (2) “Oriental”, (3) “(Hemiptera: Heteroptera: Reduviidae)”
Page 1, Paragraph 4, and lines 21: “Abstract: The two closely related oriental species of the genus Rhynocoris, R. costalis”
|
||
|
Response 1: Thank you for pointing this out. We agree with this comment. We discussed and modified the title as follows –Page 1, Paragraph 1, and lines 2,3 Title: “Comments on two controversial Oriental assassin bug species of the genus Rhynocoris (Heteroptera: Reduviidae: Harpactorinae), with the description of R. minutus sp. nov. from China]”
Thank you for pointing this out. The first letter of "Oriental" is capitalized –Page 1, Paragraph 4, and lines 21 “Abstract: The two closely related Oriental species of the genus Rhynocoris, R. costalis”
|
||
|
Comments 2: “Hemiptera: Heteroptera: Reduviidae”, “occurring”–Page 1, Paragraph 6, and lines 37-40: “The assassin bug genus Rhynocoris Hahn, 1834 (Hemiptera: Reduviidae:Harpactorinae) is a species-rich group of about 150 described species distributed mainly in the Eastern Hemisphere, with only two occur in North America [1].”
|
||
|
Response 2: Thank you for pointing these out. We have revised this sentences. Rhynocoris ventrais is widespread across the USA, while the Palaearctic species Rhynocoris eucospilus is known from Sitka, AK.–Page 1, Paragraph 6, and lines 37-40.
“The assassin bug genus Rhynocoris Hahn, 1834 (Hemiptera: Heteroptera: Reduviidae: Harpactorinae) is a species-rich group of about 150 described species distributed mainly in the Eastern Hemisphere, with only two species occurring in North America [1].”
|
||
|
Comments 3: How many of them are described by Miller or Ambrose and therefore doubtful? :-) – Page 2, Paragraph 1, and lines 41
|
||
|
Response 3: The distribution and species number of the genus Rhynocoris are as follows: Africa (82), Palaearctic region (45), Eastern Region (30), and Neoearctic region (2). Some of them is subspecies and varieties. Miller described 19 Rhynocoris species. Ambrose didn’t published new species of Rhynocoris.–Page 2, Paragraph 1, and lines 41
|
||
|
Comments 4: Page 2, Paragraph 1, and lines 41: ...which reaching neither the collar anteriorly nor the transverse constriction... Page 2, Paragraph 2, and lines 45: Harpactor in previous literatures until the publishment of the catalogues of Page 2, Paragraph 2, and lines 61: smaller in size. The key is provided for the identification of the Rhynocoris species
|
||
|
Response 4: Thank you for pointing these wrong words out. Page 2, Paragraph 1, and lines 41: “...which reaches neither the collar anteriorly nor the transverse constriction...” Page 2, Paragraph 2, and lines 45: “...Harpactor in previous literature until the publication of the catalogues of... ” Page 2, Paragraph 2, and lines 61: ...is smaller in size. A key is provided for the identification of the Rhynocoris species...
|
||
|
Comments 5: References for the published records? “Distribution. China (Sichuan, Fujian, Taiwan, Guangdong, Guangxi, Yunnan, 238 Xizang, Hainan); Myanmar; India; Sri Lanka; Vietnam (new record); Bangladesh; 239 Cambodia (new record); Malaysia; Indonesia (Java, Sumatra).”–Page 7, Paragraph 4, and lines 237-239
“Distribution. China (Guizhou, Yunnan); Japan; Vietnam; India; Sri Lanka; Indonesia; Philippines.”–Page 13, Paragraph 2, and lines 351-352
|
||
|
Response 5: Thank you for pointing these out. References have been added to the distribution of species. Page 7, Paragraph 4, and lines 237-239. “Distribution. China (Sichuan, Fujian, Taiwan, Guangdong, Guangxi, Yunnan, Xizang, Hainan) [3]; Myanmar [3]; India[3]; Sri Lanka [3]; Vietnam (new record); Bangladesh [25]; Cambodia (new record); Malaysia [3]; Indonesia (Java, Sumatra) [3].”
Page 13, Paragraph 2, and lines 351-352: “Distribution. China (Guizhou, Yunnan, Fujian, Hainan)[11]; Vietnam[11]; India[29]; Sri Lanka[31]; Indonesia[31]; Philippines[23]; Laos [11]; Myanmar[11]; Thailand [11].”
|
||
|
Comments 6: Page 7, Paragraph 2, and lines 246: “controversial about their distribution and taxonomy in Chinese literatures [3]. However,” |
||
|
Response 6: Thank you for pointing these wrong words out. Page 7, Paragraph 2, and lines 246: “confused concerning their distribution and taxonomy in Chinese literature [3]. However, “
|
||
|
Comments 7: Use traditional ZMUC instead throughout the text. Page 13, line 355: female specimens deposited in the collection of NHMD are recognized as syntypes of Page 2, line 68: NHMD Natural History Museum of Denmark, Copenhagen, Denmark Page 9, line 265: Reduvius fuscipes Fabricius, 1787: 312 [29]. Syntypes (2♀): “India Orientalis”, NHMD Page 9, line 277: Type material examined. Syntypes: 2♀, “India Orientalis” (NHMD)(Figure 4). Page 10, line 305: dorsal view; (B, E) lateral view; (C, F) ventral view (©NHMD, photographed by Sree G. Page 21, line 607: (NHMD) and Gunvi Lindberg (NHRS) for providing information and photographs of the type
|
||
|
Response 7: Thank you for pointing these out. All is revised. Page 13, line 355: female specimens deposited in the collection of ZMUC are recognized as syntypes of Page 2, line 68: ZMUC Zoological Museum of University of Copenhagen, Copenhagen, Denmark Page 9, line 265: Reduvius fuscipes Fabricius, 1787: 312 [29]. Syntypes (2♀): “India Orientalis”, ZMUC Page 9, line 277: Type material examined. Syntypes: 2♀, “India Orientalis” (ZMUC)(Figure 4). Page 10, line 305: dorsal view; (B, E) lateral view; (C, F) ventral view (©ZMUC, photographed by Sree G. Page 21, line 607: (ZMUC) and Gunvi Lindberg (NHRS) for providing information and photographs of the type
|
||
|
Comments 8: the term is correctly phallosoma, soma in Greek means "body", quite common term in anatomical and morphological terms-Page 15, and line 427 “sclerite short and only covering basal 3/5 of phallosome; lateral arms of dorsal”
|
||
|
Response 8: Thank you for pointing this out. All the “phallosome” and “endosome” are revised as “phallosoma” and “endosoma” throughout the text.
|
||
|
Comments 9: the Latin adjective minutus-Page 17 and line 454 “ Etymology. The specific epithet is derived from Latin minutus (meaning small or smaller), referring here to the relatively small body size of this species.”
|
||
|
Response 9: Thank you for pointing this out. “Etymology. The specific epithet is derived from the Latin adjective minutus (meaning small or smaller), referring here to the relatively small body size of this species.“
|
||
|
Comments 10: “amount of eggs”, “nymphal instars”-Page 17, and lines 465-467 ”of ovipositions, oviposition amount and hatching rate [37]. There are five instars of nymphs (Figure 9B–F). Adults occur almost throughout the year, whereas the peak period is from July to September (Figure 9G–I). In Nanxiong twon, Guangdong province“
|
||
|
Response 10: Thank you for pointing these wrong words out.-Page 17, and lines 465-467 “of ovipositions, amount of eggs and hatching rate [37]. There are five nymphal instars (Figure 9B–F). Adults occur almost throughout the year, whereas the peak period is from July to September (Figure 9G–I). In Nanxiong town, Guangdong province”
|
||
|
Comments 11: Which groups the prey taxa belong? Page 18, lines 506, 507: “Feeding habit. This species has a wide diet and can prey on a variety of farmland pests, such as Spodoptera litura, Spodoptera frugiperda, Agrotis ipsilon, Myzus persicae and Helicoverpa assulta [38–40].” Page 18, lines 547-550: “Feeding habit.This species has a relatively wide diet and can prey on approximately 42 species of agricultural pests, including Spodoptera litura, Spodoptera frugiperda, Earias fabia, Helicoverpa armigera, Dysdercus cingulatus, Epilachna vigintioctopunctata and Nilaparvata lugens [14, 42–46].” Page 18, lines 581-584: “agricultural production, such as Spodoptera litura, Spodoptera frugiperda, Agrotis ipsilon, Myzus persicae, Helicoverpa armigera, Epilachna vigintioctopunctata, and Nilaparvata lugens [14, 38, 43–46]. This means that they can be released in fields of various crops, such as cotton, brinjal, rice, groundnut, and tobacco [38, 43, 46, 51, 52]. Although they have a”
|
||
|
Response 11: Thank you for pointing these out.
Page 18, lines 506, 507: “Feeding habit. This species has a wide diet and can prey on a variety of farmland pests, such as Spodoptera litura (Fabricius, 1775) (Lepidoptera: Noctuidae)[38, 39], Agrotis ipsilon (Walker, 1865) (Lepidoptera: Noctuidae)[39], Myzus persicae (Sulzer, 1776) (Hemiptera: Aphididae)[38, 39], Helicoverpa assulta (Guenée, 1852) (Lepidoptera: Noctuidae) [38, 39], Spodoptera frugiperda (J.E.Smith, 1797) (Lepidoptera: Noctuidae)[40].”
Page 18, lines 547-550: “Feeding habit. This species has a relatively wide diet and can prey on approximately 42 species of agricultural pests, including Spodoptera frugiperda J.E. Smith, 1797 (Lepidoptera: Noctuidae)[14], Earias fabia (Stoll, 1782) (Lepidoptera: Nolidae)[42], Helicoverpa armigera (Hübner, 1808) (Lepidoptera: Noctuidae)[42], Dysdercus cingulatus (Fabricius, 1775) (Hemiptera: Pyrrhocoridae)[14, 43], Spodoptera litura (Fabricius, 1775) (Lepidoptera: Noctuidae)[14, 44], Nilaparvata lugens (Stål, 1854) (Hemiptera: Delphacidae) [14, 45], Epilachna vigintioctopunctata (Fabricius, 1775) (Coleoptera: Coccinellidae)[46], and so on.”
Page 18, lines 581-584: “agricultural production, such as Spodoptera litura (Fabricius, 1775) (Lepidoptera: Noctuidae), Spodoptera frugiperda (J.E.Smith, 1797) (Lepidoptera: Noctuidae), Agrotis ipsilon (Walker, 1865) (Lepidoptera: Noctuidae), Myzus persicae (Sulzer, 1776) (Hemiptera: Aphididae), Helicoverpa armigera (Hübner, 1808) (Lepidoptera: Noctuidae), Epilachna vigintioctopunctata (Fabricius, 1775) (Coleoptera : Coccinellidae), and Nilaparvata lugens (Stål, 1854) (Hemiptera: Delphacidae) [14, 38, 43–46]. This means that they can be released in fields of various crops, such as Gossypium hirsutum Linn. (cotton), Solanum melongena Linn. (brinjal), Oryza sativa Linn. (rice), Arachis hypogaea Linn. (groundnut), and Nicotiana tabacum Linn.(tobacco) [38, 43, 46, 51, 52]. Although they have a”
|
||
|
Comments 11: Summarize differences for larvae of R. costalis and R. fuscipes–Page 18, lines 506, 507
|
||
|
Response 11: Thank you for pointing these out. –Page 18, line 521. “The nymphs is orange in R. fuscipes and orange-red in R. costalis, but the color markings are very similar for the nymphs of two species ”
|
||
|
4. Response to Comments on the Quality of English Language |
||
|
Point 1: The English require a final reading by a native speaker. The English could be improved to more clearly express the research. The English require a final reading by a native speaker. |
||
|
Response 1: we will try to improve the quality of English language. |
||
|
5. Additional clarifications |
||
|
|
||

Reviewer 3 Report
Comments and Suggestions for Authors
This manuscript advances our knowledge of Rhynocoris by redescribing two species, describing one new species, providing an identification key to Chinese species, and including high-quality live, habitus, and structural images. The authors also contribute valuable observations on the biology of these species. I recommend publication after major revisions to improve the clarity, grammar, structure, terminology, and align the Discussion more closely with the data presented. I provided line-by-line comments and suggestions in the attached Word document, and summarize the main issues below:
Materials and Methods:
The manuscript would benefit from clarification on how live specimens were collected and handled, and whether feeding observations reflect lab trials or literature reports. These details are important for reproducibility and for interpreting the biological observations presented.
Species Descriptions & Taxonomic Key:
The descriptions are difficult to follow due to overly long or unpunctuated sentences, inconsistent phrasing, or ambiguous terms (e.g., “rough,” “markings”). For the coloration section, readability could be improved by breaking up sentences and grouping information by body region. The key would benefit from consistent phrasing (e.g., avoid using pronouns like “its”) and more details for character states (e.g., “large anterior lobe” is vague).
Discussion:
The Discussion includes claims about agricultural biocontrol that are not directly supported by the current study. I suggest focusing the discussion more on the study’s core findings (life history, morphology) and either omitting or briefly contextualizing the applied relevance with appropriate caveats.
Grammar and English usage:
Multiple phrases require editing for grammar and fluency. Several suggestions are noted in the attached file.

While generally understandable, I recommend careful language revision as the manuscript includes frequent issues with grammar, word choice, and phrasing that affect readability. I provide some specific examples in the attached Word file.
Author Response
For research article
|
Response to Reviewer 3 Comments
|
||
|
1. Summary |
|
|
|
Thank you very much for taking the time to review this manuscript. Please find the detailed responses below and the corresponding revisions in track changes in the re-submitted files.
|
||
|
2. Questions for General Evaluation |
Reviewer’s Evaluation |
Response and Revisions |
|
Does the introduction provide sufficient background and include all relevant references? |
Can be improved |
Thank you |
|
Is the research design appropriate? |
Yes |
Thank you |
|
Are the methods adequately described? |
Can be improved |
Thank you |
|
Are the results clearly presented? |
Can be improved |
Thank you |
|
Are the conclusions supported by the results? |
Must be improved |
Thank you |
|
Are all figures and tables clear and well-presented? |
Yes |
Thank you |
|
3. Point-by-point response to Comments and Suggestions for Authors |
||
|
Comment 1: This manuscript advances our knowledge of Rhynocoris by redescribing two species, describing one new species, providing an identification key to Chinese species, and including high-quality live, habitus, and structural images. The authors also contribute valuable observations on the biology of these species. I recommend publication after major revisions to improve the clarity, grammar, structure, terminology, and align the Discussion more closely with the data presented. I provided line-by-line comments and suggestions in the attached Word document, and summarize the main issues below:
|
||
|
Response 1: Thank you very much. We will try to improve the quality of English language.
|
||
|
Comment 2: Materials and Methods: The manuscript would benefit from clarification on how live specimens were collected and handled, and whether feeding observations reflect lab trials or literature reports. These details are important for reproducibility and for interpreting the biological observations presented.
|
||
|
Response 2: “These two species are common natural enemies in pest control in South China and can be directly collected by hand from plants in agricultural ecosystems.” “The reduviids were all reared under laboratory conditions (26±2 ℃, natural light) and fed on larvae of yellow mealworm, Tenebrio molitor Linnaeus, 1758 (Coleoptera: Tenebrionidae). ” “In laboratory, they usually lay their eggs on rough surfaces of the egg cartons (Figure 9A). ” “The eggs and 1–5 instars nymphs were described, and photographed on artificially placed leaves in a laboratory setting.”
|
||
|
Comment 3: Species Descriptions & Taxonomic Key: The descriptions are difficult to follow due to overly long or unpunctuated sentences, inconsistent phrasing, or ambiguous terms (e.g., “rough,” “markings”). For the coloration section, readability could be improved by breaking up sentences and grouping information by body region. The key would benefit from consistent phrasing (e.g., avoid using pronouns like “its”) and more details for character states (e.g., “large anterior lobe” is vague).
|
||
|
Response 3: Thank you for pointing these out. We agree with this comment. We revise these issues throughout the text.
|
||
|
Comment 4: Discussion: The Discussion includes claims about agricultural biocontrol that are not directly supported by the current study. I suggest focusing the discussion more on the study’s core findings (life history, morphology) and either omitting or briefly contextualizing the applied relevance with appropriate caveats.
|
||
|
Response 4: Thank you for pointing these out. We agree with this comment. We revise section of discussion.
|
||
|
Comment 5: Grammar and English usage: Multiple phrases require editing for grammar and fluency. Several suggestions are noted in the attached file.
|
||
|
Response 5: Thank you very much. We will try to improve the quality of English language.
|
||
|
Comment 6: INTRODUCTION Line 8 – “(H.Y. L.)”: Is the spacing in between the Y and L intentional? |
||
|
Response 6: Thank you for pointing this out. We have revised it. “1Beijing 100193, China; huaiyuliu0103@163.com (H.Y.L.); insectchen625@126.com (Z.C.); cqxiong_hy@126.com (H.Y.X.); zhaoyangchen@cau.edu.cn (Z.Y.C.); tigerleecau@hotmail.com (H.L.); caiwz@cau.edu.cn (W.C.) 2Key Laboratory of Environment Change and Resources Use in Beibu Gulf (Ministry of Education), Nanning Normal University, Nanning 530001, China; zpyayjl@126.com (P.Z.) * Correspondence: zpyayjl@126.com (P.Z.); caiwz@cau.edu.cn (W.C.); Tel.: +0086-010-62732885.”
|
||
|
Comment 7: Line 15 – “… , and has been proved to play a significant role”: Style issue: consider simplifying by deleting “been”
|
||
|
Response 1: Thank you for pointing this out. We have deleted it.
|
||
|
Comment 7: Lines 16/17 – “Based on the examination of type specimens and morphological studies,”: Unclear phrase, as written it sounds like you are examining morphological studies, rather than conducting your own. Consider rephrasing to clarify that you are examining type specimens and morphology.
|
||
|
Response 8: Thank you for pointing this out. We have revised it. “Based on the examination of type specimens and morphology, two Rhynocoris species widely distributed in the Oriental Region,...”
|
||
|
Comment 8: Line 23 – “…exhibit remarkable morphological similarity, especially the red and black color patterns.” Clarify where this is observed. It can be as general as "...particularly in their overall red and black body coloration” to something more descriptive, e.g., “particularly in their overall red and black body coloration, including the head, thorax, and abdomen.”
|
||
|
Response 9: Thank you for pointing this out. We have revised it. “exhibit remarkable morphological similarity, particularly in their overall red and black body coloration, including the head, thorax, and abdomen.”
|
||
|
Comment 9: Line 24 – “…collected from various localities” Vague. Various localities in which region(s)?
|
||
|
Response 9: Thank you for pointing this out. We have revised it. “Based on the examination of type specimens and non-type material collected from various localities of South China......”
|
||
|
Comment 10: Lines 27/28 – “… , most studies on the taxonomic, biological and biocontrol studies under the name …” The word "studies" may be overused in this sentence. Consider something like, "most taxonomic, biological, and biocontrol studies conducted under the name..." |
||
|
Response 10: Thank you for pointing this out. We have revised it. “..., most taxonomic, biological and biocontrol studies conducted under the name Harpactor fuscipes or Rhynocoris fuscipes should be instead attributed to R. costalis. ”
|
||
|
Comment 11: Line 39 – “…with only two occur in North America.” "only two occur" --> "...only two species occurring" |
||
|
Response 11: Thank you for pointing this out. We have revised it. “...with only two species occurring in North America...”
|
||
|
Comment 12: Lines 39/40 – “Most of them were found in the Ethiopian Region (>80 spp.),” Ambiguous phrasing. Most what? Found how? Consider clarifying, for example, "Most species have been recorded from..." or "known from" or "the majority of species are distributed in..." |
||
|
Response 12: Thank you for pointing this out. We have revised it. “The majority of species are distributed in the Ethiopian Region (>80 spp.), ”
|
||
|
Comment 13: Line 43 – “the relatively large anterior lobe of the pronotum,” Ambiguous phrasing. "Large" in what way? Wide, long, tall? |
||
|
Response 13: Thank you for pointing this out. We have deleted “the relatively large”. “the anterior pronotal lobe...“
|
||
|
Comment 14: Line 45 – “which reaching neither the collar anteriorly” Grammar. reaching --> reaches
|
||
|
Response 14: Thank you for pointing this out. We have revised it. “which reaches neither the collar anteriorly”
|
||
|
Comment 15: Line 46 – “can be found in varies vegetated” varies --> various
|
||
|
Response 15: Thank you for pointing this out. We have revised it. “...can be found in various vegetated...”
|
||
|
Comment 16: Lines 54 & 246 – “literatures” Grammar. Literatures --> literature
|
||
|
Response 16: Thank you for pointing this out. We have revised it. “...Harpactor in previous literature until the publication of the catalogues of...” |
||
|
Comment 17: Line 54 – “publishment” Word choice. Outdated term, consider using "publication."
|
||
|
Response 17: Thank you for pointing this out. We have revised it. “...Harpactor in previous literature until the publication of the catalogues of...”
|
||
|
Comment 18: Lines 55/56 – “Due to the identification confusion” Awkwardly phrased. Consider rephrasing to something like “Due to misidentification.”
|
||
|
Response 18: Thank you for pointing this out. We have revised it. “...Due to misidentification ...”
|
||
|
Comment 19: Lines 52-52 – Somewhere in the last paragraph of the introduction, highlight that this is a two part study, (1) conducting the taxonomy and (2) providing insights into the biology of an important group of reduviids where there is much to figure out still.
|
||
|
Response 19: Thank you for pointing this out. We have revised it. “we conducted a reexamination of the type specimens and redescribed these two taxonomically controversial species. Furthmore, the biology of these two species was reported for further providing insights into the important group of reduviids.”
|
||
|
Comment 20: METHODS Line 79 – “microscope […] for dissected body parts.” Does this refer exclusively to genitalia or were other structures also dissected and examined?
|
||
|
Response 20: Thank you for pointing this out. We have revised it. “microscope (Olympus Inc., Tokyo, Japan) for dissected genitalia.”
|
||
|
Comment 21: Lines 85-87 – “The live samples of Rhynocoris costalis (Stål, 1867) were collected from Shaoguan, Guangdong Province, China. The samples of Rhynocoris fuscipes (Fabricius, 1787) were from Buga Town, Zhaotong, Yunnan Province, China.” Consider including a brief description of collection methods as this provides useful ecological context and supports reproducibility. E.g., whether specimens were collected by hand, with light traps, sweep nets, etc.
|
||
|
Response 21: Thank you for pointing this out. We have revised it. “microscope (Olympus Inc., Tokyo, Japan) for dissected genitalia.”
|
||
|
Comment 22: Line 90 – “The eggs and 1–5 instars nymphs were described and illustrated.” Instead of “illustrated”, photographed might be more accurate. It would also be helpful to clarify whether the live images were taken in a laboratory setting (e.g., on artificially placed leaves or neutral backgrounds) or on natural vegetation. This information would help readers interpret the context and realism of the images. |
||
|
Response 22: Thank you for pointing this out. We have revised it. “The eggs and 1–5 instars nymphs were described, and photographed on artificially placed leaves in a laboratory setting.”
|
||
|
Comment 23: RESULTS Line 107 – “sepceis” Spelling: sepceis -> species
|
||
|
Response 23: Thank you for pointing this out. We have revised it. “Note: Hsiao and Ren in 1981 keyed 10 species is and 2 subspecies of Rhynocoris in China ”
|
||
|
Comment 24: KEY Lines 112-115 – “1. Body small-sized, 12–14 mm or even smaller; second antennal segment generally longer than third segment...2 -. Body generally large-sized, typically more than 14 mm; second antennal segment generally shorter than, equal to, or slightly longer than the third antennal segment...7” The terms “small-sized” and “generally large-sized” could be revised for clarity, for example, using “relatively short” and “relatively long” which are commonly used in morphological descriptions. The phrase “Body small-sized, 12–14 mm or even smaller” is a bit unclear. If specimens can be smaller than 12 mm, it may be clearer to state “up to 14 mm long” or to provide the observed lower limit of the body length range. As written, “shorter than, equal to, or slightly longer than” the third segment covers a broad and overlapping range that may not contrast clearly with “generally longer than” in the first couplet. It may be helpful to define how much longer qualifies as “longer” versus “slightly longer,” or to revise the character state to one that is more discrete or measurable. For example, "second antennal segment distinctly longer than third” and the second as “second antennal segment subequal to third,” or adjust the wording as needed to better reflect the observed morphological variation.
|
||
|
Response 24: Thank you for pointing this out. Thank you, reviewer 3, for your meticulous revision and careful guidance of our manuscript. I tried my best to fully understand the suggestions of reviewer 3. I was very worried that I didn't fully grasp them, but I did my best to make revisions. I really hope that all the revisions have become reasonable. After the modification, try to use as few “small-sized” and “generally large-sized” values as possible. The key revised is as followed: “ The key to the Chinese species in the genus Rhynocoris Hahn, 1834 1. Anterior pronotal lobe with dense appressed setae and glabrous arc-shaped sculptures...2 -. Anterior pronotal lobe smooth, without dense appressed setae and glabrous sculptures...3 2. Body relatively long and robust, over 14.5 mm in male and 16.5 mm in female in length; anterior pronotal lobe with deep glabrous sculptures...Rhynocoris incertis (Distant, 1903) -. Body relatively short and slender, less than 14.0 mm in male and 16.0 mm in female in length; anterior pronotal lobe with shallow glabrous sculptures...Rhynocoris marginellus (Fabricius, 1803) 3. Second antennal segment longer than third segment...4 -. Second antennal segment shorter than third segment...9
Comment 25: Line 116 – “black regular transversal intersegmental stripes. Does “regular” refer to spacing, width, or some other aspect? Consider specifying whether the stripes are regularly spaced, uniform in width, or whatever it may be Response 25: we have deleted “regular”. 4. Abdomen ventrally red to orange with black transversal intersegmental stripes...Rhynocoris minutus sp. nov.
-. Abdomen ventrally black, or red to orange with irregular black markings ...5 5. Connexivum unicolor, reddish...Rhynocoris rubromarginatus (Jakovlev, 1893) -. Connexivum bicolor, basal half of each segment black, apical half red...6
Comment 26: Line 121 – “ Femora with distinct wide red annular markings” Consider specifying "All femora..." if markings are present on all pairs Comment 26: Line 122 – “ Femora with obscure annular markings or completely black” Are the obscure annular markings a specific color (e.g., blackish red or another shade)? Clarifying this could help distinguish between obscure markings and absent markings. Response 26: we have deleted “distinct ”. Fomora- >All femora; obscure ->vague “ 6. All femora with wide red annular markings...7 -. All femora with vague annular markings or completely black...8 ” Comment 27: Line 123 & 125 – “Basal part of each femur with a wide red annular marking...Basal and middle parts of each femur with two wide red annular markings” For this couplet, consider adding "single" for parallel structure. For example, something like: “All/Each femur with single red annular marking basally. - Each femur with two red annular markings, basally and medially.”
Response 27: “ 7. Each femur with single red annular marking basally...Rhynocoris altaicus Kiritshenko, 1926 -. Each femur with two red annular markings, basally and medially..Rhynocoris dauricus Kiritshenko, 1926 ”
Comment 28: Line 129 – “exterior margin” Ambiguous phrasing. Consider clarifying, i.e., posterior, lateral, anterior margin
Response 28: Thank you for pointing this out. We have revised it. 8. Pronotum completely black; all femora black, with pale annular marking basally and medially...Rhynocoris leucospilus (Stål, 1859) -. Pronotum black, sometimes lateral margin of lateral pronotal angle red; all femora completely black...Rhynocoris sibiricus (Jakovlev, 1893)
9. Connexivum black; body entirely black, except red coxae and trochanters...Rhynocoris reuteri (Distant, 1879) -. Connexivum bicolor, red and black; body red with black markings...10 10. Femora red, with irregular black annular markings...11 -. Femora black, without annular markings...12 11. Body relatively long, more than 18 mm in length; ventral surface of abdomen completely black; second antennal segment slightly longer than or approximately equal to third segment...Rhynocoris monticola (Oshanin, 1871) -. Body relatively short, less than 18 mm in length; ventral surface of abdomen red, with black longitudinal markings in middle part and on both sides; second antennal segment noticeably shorter than third segment...Rhynocoris iracundus (Poda, 1761)
Comment 29: Line 148 – “ Femora black, its ventral surface” You don’t need pronouns like “its” in taxonomic keys. Comment 29: Lines 145 & 151 – “ventral surface of abdomen red” and “abdominal ventral surface red” Consider revising for parallel structure.
Response 29: Thank you for pointing this out. We have revised it. “ 12. Femora black, ventral surface with a series of white spots; sterna of abdomen red, with yellowish-white and black intersegmental stripes...Rhynocoris costalis (Stål, 1867) -. Femora completely black; sterna of abdomen red, with black intersegmental stripes...Rhynocoris fuscipes (Fabricius, 1787) ” |
||
|
Comment 30: DESCRIPTIONS Lines 179-188 & others – “Transversal stripe between eyes located before transversal constriction of head above, postocular part, anterior lateral angle of pronotum, anterior pronotal lobe, most of posterior pronotal lobe (except transversal constriction and middle part of anterior area black), propleuron (except anterior angle and a large spot in middle part black), mesopleuron and metapleuron (except irregular markings of upper margin), thoracic sterna (except spots of middle part black), apical half of scutellum, most of corium (except inner sides black), each connexival segment (sometimes basal part and external margin black), coxal cavities, coxae, trochanters, ventral surface of abdomen (except intersegmnetal stripes and large spots of two lateral sides black and yellowish- white) red. The descriptions currently list many body parts and end with the unique trait. This structure is difficult to follow due to the long, unpunctuated sentence structure. I recommend starting with the color or breaking this into multiple, shorter sentences grouped by body region with consistent phrasing and parallel structure.
Response 30: Thank you for pointing this out. We have revised it. We start sentences with the color first.
|
||
|
Comment 31: Line 192 – eye color black Describing the eye color may not be necessary since it can change depending on preservation method, lighting, or life stage.
Response 31: Thank you for pointing this out. We have deleted it. |
||
|
|
||
|
Comment 32: Line 196 – “two lateral side (except white spots) ” • Confusing phrasing
|
||
|
Response 32: Thank you for pointing this out. “ lateral sides of abdominal sterna (except white spots), transversal stripe of... ”
|
||
|
Comment 33: Line 216 – “a parasol-shaped sclerotized support structure in middle part” • Having difficulty identifying this structure from the images. Does the literature have a name for this structure?
|
||
|
Response 33: Thank you for pointing this out. The structure of male genital have been marked and annotated in Figures 3, 6, 8.
|
||
|
Comment 34: Line 247 – “known natural enemies in agricultural pest control” • Cite studies.
|
||
|
Response 34: Thank you for pointing this out. “...given their significance as known natural enemies in agricultural pest control[3, 5–7, 12–14], it... |
||
|
Comment 35: Lines 322–329 • Might be an illusion, but this text appears smaller than the rest of the body.
|
||
|
Response 35: Yes, it is. Thank you for pointing this out.
|
||
|
Comment 36: Line 366 – “most noticeable difference being: In R. costalis,” No colon after a verb like "being" and lowercase "i". You might consider rephrasing to keep the colon and parallel structure or switching to a comma.
|
||
|
Response 36: Thank you for pointing this out. “...spots. The structure of the male genitalia in both species is noticeable different. In R. costalis, the central part of the dor...”
|
||
|
Comment 37: Line 391 – “abdomen beneath” Strange word choice, consider using something like ventral abdominal surface. Comment 37: Line 392 – “without intersegment stripes.” intersegment --> intersegmental
|
||
|
Response 37: Thank you for pointing this out. “...more than 11 mm in male and 14 mm in female, and the ventral abdominal surface has no transversal intersegmental stripes. ...” |
||
|
Comment 38: Line 412 – “yellowish white short appressed setae” Revise punctuation to clarify how the modifiers are grouped. E.g., yellowish-white short-appressed setae, or yellowish-white, short, appressed setae?
|
||
|
Response 38: Thank you for pointing this out. “covered with yellowish-white, short, appressed setae; legs sparsely covered with ”
|
||
|
Comment 39: Line 468 – “of China, Rhynocoris costalis” Abbreviate genus
|
||
|
Response 39: Thank you for pointing this out. “R. costalis has two generations a year, with adults beginning to overwinter from...“
|
||
|
Comment 40: Line 479: Figure 9H Is H also a female? I would recommend changing “preying” to “feeding” if it was lab-fed and provided prey. There seems to be a discrepancy between the photo, which shows the insect feeding on Mythimna separata, and the methods section, which states they were fed Tenebrio molitor. Please clarify whether both prey types were used, and under what conditions the feeding in the image occurred.
|
||
|
Response 40: Thank you for pointing this out. “...female; (H) female, feeding on larvae of the pest Mythimna separata in the laboratory; (I) female, left,...” |
||
|
Comment 41: Lines 507 & 550 – “Feeding habit. This species has a wide diet and can prey on a variety of farmland pests, such as Spodoptera litura, Spodoptera frugiperda, Agrotis ipsilon, Myzus persicae and Helicoverpa assulta [38–40].” It is unclear whether the listed prey items are based on the authors’ own observations or cited entirely from previous literature. Since this section appears under the results, readers may interpret it as new data. I recommend clarifying either in the methods (if feeding trials were conducted) or by stating explicitly here that the diet information is based on previous studies [38–40]. Consider doing the same for the next species in line 550.
|
||
|
Response 41: Thank you for pointing this out.
“Feeding habit. This species has a wide diet and can prey on a variety of farmland pests, such as Spodoptera litura (Fabricius, 1775) (Lepidoptera: Noctuidae)[38, 39], Agrotis ipsilon (Walker, 1865) (Lepidoptera: Noctuidae)[39], Myzus persicae (Sulzer, 1776) (Hemiptera: Aphididae)[38, 39], Helicoverpa assulta (Guenée, 1852) (Lepidoptera: Noctuidae) [38, 39], Spodoptera frugiperda (J.E.Smith, 1797) (Lepidoptera: Noctuidae)[40].”
“Feeding habit. This species has a relatively wide diet and can prey on approximately 42 species of agricultural pests, including Spodoptera frugiperda J.E. Smith, 1797 (Lepidoptera: Noctuidae)[14], Earias fabia (Stoll, 1782) (Lepidoptera: Nolidae)[42], Helicoverpa armigera (Hübner, 1808) (Lepidoptera: Noctuidae)[42], Dysdercus cingulatus (Fabricius, 1775) (Hemiptera: Pyrrhocoridae)[14, 43], Spodoptera litura (Fabricius, 1775) (Lepidoptera: Noctuidae)[14, 44], Nilaparvata lugens (Stål, 1854) (Hemiptera: Delphacidae) [14, 45], Epilachna vigintioctopunctata (Fabricius, 1775) (Coleoptera: Coccinellidae)[46], and so on. ”
|
||
|
Comment 42: DISCUSSION The Discussion would benefit from a more balanced perspective by acknowledging potential limitations of the study. For example, clarifying constraints in methods, sampling limitations, or uncertainties in interpreting biological observations (e.g., behavior, habitat use) would help contextualize the findings and guide future research. Including such reflections is standard in scientific writing and strengthens the credibility and depth of the discussion.
|
||
|
Response 42: Agree. We have, accordingly, changed discussion.
|
||
|
Comment 43: Lines 553-556 – “Many species of the genus Rhynocoris exhibit varied and complex color pattern variations [3]. For example, Rhynocoris marginellus shows color pattern variation on connexivum, pronotum and fore femur [27]. Putshkov [47] also used Rhynocoris persicus as an example to illustrate the complexity of intraspecific variation within the genus.” This paragraph makes an important point about intraspecific variation in Rhynocoris, but the discussion of intraspecific color variation in the cited examples are vague. To strengthen the flow and purpose of the first few sentences, I recommend slightly reframing the opening to more directly connect the general examples (R. marginellus, R. persicus) to the focal species being studied. For instance, stating up front that these examples serve to contextualize or justify the focus on R. costalis and R. fuscipes would help the reader better anticipate the relevance of the paragraph and maintain continuity. Alternatively, condensing the first few citations and leading with the focal species might streamline the paragraph. (e.g., Many species of the genus Rhynocoris exhibit varied and complex color pattern variations, as documented in species such as R. marginellus [27], R. persicus [47], and others [3].)
|
||
|
Response 43: Agree. We have, accordingly, changed discussion.
|
||
|
Comment 44: Line 556 – “…within the genus. R. costalis and …” Writing conventions recommend avoiding abbreviated genus names (e.g., R.) at the beginning of a sentence. Consider spelling out Rhynocoris when starting a new sentence. |
||
|
Response 44: Agree. We have, accordingly, changed discussion. |
||
|
Comment 45: Lines 570-571 – “…this species has a significantly smaller body size, which is presumed to be an adaptation to the high-altitude environment” The statement that smaller body size in R. minutus is “presumed to be an adaptation to the high-altitude environment” may be overstated without direct ecological or physiological evidence. I recommend rephrasing this more cautiously (e.g., “may reflect adaptation” or “could be associated with high-altitude conditions”) unless the cited references provide strong, taxon-specific support for this claim.
|
||
|
Response 45: Agree. “...native to China, this species has a significantly smaller body size, which could be associated with high-altitude conditions [48, 49]. ...”
|
||
|
Comment 46: Line 573 – “Both Rhynocoris costalis and Rhynocoris fuscipes” Generic names can be abbreviated in the middle of a sentence. Section “4.2. Effective reduviid predators in the agricultural ecosystem” The agricultural biocontrol discussion feels disconnected from the core findings of the manuscript, which focus on documenting life history and morphology of R. costalis and R. fuscipes. I recommend centering the discussion on interpreting your own observations then briefly considering how these findings may inform future applied or ecological work. E.g., how your taxonomic work will affect applied agricultural work in the first section, and the second section discussing the biology you studied. If you choose to keep a section dedicated to agricultural relevance, I suggest clearly distinguishing between the contributions of your current study and those drawn from the cited literature. Additionally, briefly acknowledging known limitations of using generalist predators like Rhynocoris in biological control (e.g., potential non-target effects, limited prey specificity, or inconsistent field performance). That said, I feel this topic may be better suited as a sentence or two in the discussion or conclusion rather than a central focus of the discussion, given the scope and goals of your study.
|
||
|
Response 46: Agree. We have, accordingly, changed discussion.
|
||
|
Comment 47: Line 588 – “Corcyra cephalonica larvae have higher survival rates and fecundity compared to those that prey on Dysdercus koenigii and Phenacoccus solenopsis.” Cite source(s).
|
||
|
Response 47: Agree. “those that prey on Dysdercus koenigii (Fabricius, 1775) and Phenacoccus solenopsis Tinsley, 1898 [52]. This indicates...”
|
||
|
4. Response to Comments on the Quality of English Language |
||
|
Point 1: The English could be improved to more clearly express the research. While generally understandable, I recommend careful language revision as the manuscript includes frequent issues with grammar, word choice, and phrasing that affect readability. I provide some specific examples in the attached Word file.
|
||
|
Response 1: We will try to improve the quality of English language.
|
||
|
5. Additional clarifications |
||
|
|
||

Round 2
Reviewer 1 Report
Comments and Suggestions for Authors
I have suggested that the authors should point the structures of the phallus, to allow their recognition by the reader. They did it, but some structures were not indicated as in Figs. 3, 6 and 8. Please, complete them.
Author Response
|
Response to Reviewer 1 Comments
|
||
|
1. Summary |
|
|
|
Thank you very much for taking the time to review this manuscript. Please find the corresponding revisions in the re-submitted files.
|
||
|
2. Questions for General Evaluation |
Reviewer’s Evaluation |
Response and Revisions |
|
Does the introduction provide sufficient background and include all relevant references? |
Yes |
Thank you |
|
Is the research design appropriate? |
Yes |
Thank you |
|
Are the methods adequately described? |
Yes |
Thank you |
|
Are the results clearly presented? |
Yes |
Thank you |
|
Are the conclusions supported by the results? |
Yes |
Thank you |
|
Are all figures and tables clear and well-presented? |
Can be improved |
Thank you |
|
3. Point-by-point response to Comments and Suggestions for Authors |
||
|
Comments 1: I have suggested that the authors should point the structures of the phallus, to allow their recognition by the reader. They did it, but some structures were not indicated as in Figs. 3, 6 and 8. Please, complete them. |
||
|
Response 1: Thank you for pointing this out. We agree with this comment. Figures 3, 6 and 8 have been added the annotations. |
||
|
4. Response to Comments on the Quality of English Language |
||
|
Point 1: The English is fine and does not require any improvement. |
||
|
Response 1:Thank you very much. |
||
|
5. Additional clarifications |
||
Reviewer 3 Report
Comments and Suggestions for Authors
Thank you for addressing all the concerns raised during the first round of reviews. The clarity of the manuscript and figures has improved significantly. I recommend acceptance after some minor revisions related to phrasing, word choice, and clarification in the Material and Methods and Discussion sections. Please refer to the attached Word document for detailed comments and suggestions.

The English has improved, but now the new parts of the Discussion section could use some attention. Some suggestions are provided in the attached file.
Author Response
For research article
|
Response to Reviewer 3 Comments
|
||
|
1. Summary |
|
|
|
Thank you very much for taking the time to review this manuscript. Please find the detailed responses below and the corresponding revisions in track changes in the re-submitted files.
|
||
|
2. Questions for General Evaluation |
Reviewer’s Evaluation |
Response and Revisions |
|
Does the introduction provide sufficient background and include all relevant references? |
Yes |
Thank you |
|
Is the research design appropriate? |
Yes |
Thank you |
|
Are the methods adequately described? |
Can be improved |
Thank you |
|
Are the results clearly presented? |
Can be improved |
Thank you |
|
Are the conclusions supported by the results? |
Yes |
Thank you |
|
Are all figures and tables clear and well-presented? |
Yes |
Thank you |
|
3. Point-by-point response to Comments and Suggestions for Authors |
||
|
Comment 1: Thank you for addressing all the concerns raised during the first round of reviews. The clarity of the manuscript and figures has improved significantly. I recommend acceptance after some minor revisions related to phrasing, word choice, and clarification in the Material and Methods and Discussion sections. Please refer to the attached Word document for detailed comments and suggestions. |
||
|
Response 1: Thank you for pointing these out. Thank you for your suggestions. We sincerely appreciate you.
|
||
|
Comment 2: SIMPLE SUMMARY Line 17—18: “Based on the examination of type specimens and morphology, …” Redundant: examining type specimens involves examining morphology. If you conducted a different kind of morphological analysis (e.g., phylogenetic or morphometric), then it would be worth specifying, otherwise, consider simplifying. |
||
|
Response 2: Agree. Thank you for pointing it out. “Based on the examination of type specimens, ...”
|
||
|
Comment 3: INTRODUCTION Line 63: “these two species was reported for further providing insights” Grammar (was were) and awkward phrasing (for further providing to provide further). |
||
|
Response 3: Agree. Thank you for pointing it out. “Furthmore, the biological information of these two species was reported to provide further insights into the important group of reduviids.”
|
||
|
Comment 4: MATERIAL AND METHODS Line 91: “These two species are common natural enemies in pest control in South China and can be directly collected by hand from plants in agricultural ecosystems.” Thanks for clarifying the material and methods for the “Biological Study” section. However, this particular sentence reads more like a fact or observation than a description of the collection methods used for this study. Rephrase to clarify whether you collected the specimens by hand in agricultural fields. If a laboratory colony was used, include how many generations or how long the colony had been maintained for (e.g., age or rearing duration) at the time of observation to contextualize the biology of these specimens and ensure reproducibility. |
||
|
Response 4: Agree. Thank you for pointing it out. “These two species are common natural enemies in pest control in South China. We collected these specimens by hand in agricultural fields. ...”
|
||
|
Comment 5: RESULTS Great job addressing consistency and clarity in the key. Just some minor clarification questions: Lines 134 & 158: “irregular black markings” · What is meant by irregular here? Shape, size, spacing, …? |
||
|
Response 5: Thank you for pointing it out. We have revised it. -. Abdomen ventrally black, or red to orange without black transversal intersegmental stripes ...5”
|
||
|
Comment 6: Line 138: “vague annular markings” Are they faintly red annular markings? |
||
|
Response 6: Thank you for pointing it out. We have revised it. It is obscure edge, brownish. “-. All femora with brownish annular markings, or completely black...8”
|
||
|
Comment 7: Line 145: “pale annular marking” · I interpret pale as a yellowish and/or white color, correct? |
||
|
Response 7: Agree. “8.Pronotum completely black; all femora black, with reddish brown to dark brown annular marking basally and medially...Rhynocoris leucospilus (Stål, 1859)”
|
||
|
Comment 8: Line 181: space in “(NHRS)(Figure 1)” “(NHRS) (Figure 1)” |
||
|
Response 8: Revised. “...(NHRS) (Figure 1)...”
|
||
|
Comment 9: Line 199: “Transversal stripe between eyes located before…” · Do you mean anterior to head constriction? Consider using anatomical terminology. |
||
|
Response 9: Thank you for pointing this out. “...Transversal stripe anterior to head constriction,...”
|
||
|
Comment 10: Lines 199, 208, 211: “reddish region”, “yellowish-white region”, etc… · The term “region” feels strange in this context considering that these groupings are based on coloration rather than anatomical regions. Consider something like “reddish-colored structures.” |
||
|
Response 10: Agree. Thank you for pointing this out. “Reddish-colored structures: Transversal stripe anterior to head constriction, ...”
|
||
|
Comment 11: Lines 207, 210, 218, etc. “[body parts] red.” “[body parts] yellowish- white.” Because you are now clarifying that each paragraph is referring to a certain color (e.g., section 1 = reddish), I don’t think you necessarily need to end each paragraph with the color again. |
||
|
Response 11: Agree. The color at the end of the paragraph was deleted.
|
||
|
Comment 12: Line 248: Measurements…interocular space If you want to use reduviid terminology, interocular space synthlipsis. |
||
|
Response 12: Thank you for pointing this out. “length of synthlipsis 1.28–1.35 / 1.32–1.37; interocellar space 0.71–0.75 / 0.70–0.79; ” “length of synthlipsis 1.27–1.41 / 1.31–1.41; interocellar space 0.65–0.76 / 0.65–0.73;” “length of synthlipsis 1.14–1.17 / 1.12–1.35 (1.14); interocellar space 0.55–0.57 / 0.55–0.66 (0.57); ”
|
||
|
Comment 13: Line 266: “when viewed from above” When viewed dorsally? |
||
|
Response 13: “...when viewed dorsally...”
|
||
|
Comment 14: ·Line 270: Remarks section · You mention that it is essential to clarify the identities of R. fuscipes and R. costalis. If you want to strengthen this goal, consider adding adedicated diagnosis or rediagnosis section for these species. The current Remarks section for R. costalis includes a helpful comparison starting on line 279 (“The yellowish-white stripes on the ventral surfaces of the femora and abdomen are characteristic for the species, which are lacking in R. fuscipes”), but this information is somewhat buried. |
||
|
Response 14: Agree. In the Remark region in R. fuscipes, all the different characters between these two species have been comparaed.
“The yellowish-white stripes on the ventral surfaces of the femora and abdomen are characteristic for the species, which are lacking in R. fuscipes (see below in Remark region of R. fuscipes).”
|
||
|
Comment 15: Line 281–286: “The structure of the male genitalia between these two species is also different, and complex and unique with the specialized structure of dorsal phallothecal sclerite, and “W”-shaped or umbrellashaped structure of endosoma among the members of the reduviid subfamily Harpactorinae (vs. In most of harpactorines, the dorsal phallothecal sclerite generally is a flat, nearly elliptical layer of sclerotized plate, and the apical part of endosoma is armed with many spines).” Consider rearranging information and clarifying what is different between the two species if that is the goal here. E.g., “In most harpactorines, the dorsal phallothecal sclerite is generally a flat, nearly elliptical layer of sclerotized plate, and the apical part of the endosoma is armed with many spines. The structure of the male genitalia of R. costalis and R. fuscipes is complex and unique among members of the reduviid subfamily Harpactorinae, yet different among the two species. In X species, the specialized structure of the dorsal phallothecal sclerite and “W”-shaped or umbrella-shaped structure of the endosoma is…” Line 282: “is also different, and complex and unique” · Redundant. Consider condensing, e.g., “are complex and unique” or “is also different, complex, and unique”. Line 283: “phallothecal sclerite, and “W”-shaped” no comma after sclerite |
||
|
Response 15: Agree. Thank you for your suggestion. “ In most harpactorines, the dorsal phallothecal sclerite is generally a flat, nearly elliptical layer of sclerotized plate, and the apical part of the endosoma is armed with many spines. The structure of the male genitalia of R. costalis and R. fuscipes is complex and unique among members of the reduviid subfamily Harpactorinae, yet different among the two species. In R. costalis, the specialized structure of the dorsal phallothecal sclerite is “W”-shaped, but umbrella-shaped in R. fuscipes.”
|
||
|
Comment 16: Line 401: “…(vs. in R. fuscipes…)” · I don’t recommend hiding this helpful information in parentheses, as parentheses often get glanced over. Lines 392–406 · Great job differentiating the two species, perfect for the diagnosis/rediagnosis section if you decide to include. |
||
|
Response 16: Agree. The parentheses have been removed.
|
||
|
Comment 17: Lines 288–289 & 407–408 · I appreciate the inclusion of intraspecific variation in the revised Remarks section. It strengthens the species diagnoses by highlighting important variability to accurately identify species. |
||
|
Response 17: Thank you.
|
||
|
Comment 18: Line 500: “One of female 500 adults in the experiment laid 8 eggs at one time and formed an egg mass.” “One of female adults” à “One of the adult females” Was this observation based on a single individual or multiple females? If multiple, was 8 eggs the minimum, average, or maximum number laid in a single event? |
||
|
Response 18: We revised it. We have not recorded in detail the number of male and female individuals. We have provided as much information as possible. A lot of biological information still needs to be observed.
“We recorded the images of the complete life history of Rhynocoris costalis (Stål, 1867) based on more than 20 live specimens collected from Shaoguan, Guangdong, China. One of the adult females in the experiment laid 8 eggs at one time and formed an egg mass. ”
|
||
|
Comment 19: Line 504: “… time of ovipositions,” · Ambiguous. Time of day or time of year/season? Clarify with something like “seasonal timing of oviposition” · ovipositions à ovipositionLine 505: “amount of eggs and hatching rate” · Comma after eggs |
||
|
Response 19: We have revised them. “such as pre-oviposition period, oviposition duration, frequency of oviposition, the fecundity per female, and hatching rate ”
|
||
|
Comment 20: Line 541: “nymphs are gregarious, and later disperse” · If you have the information, how much later? |
||
|
Response 20: After the 2nd instar. “Life habit. Newly hatched nymphs are gregarious, and later disperse to move independently after the 2nd instar. ”
|
||
|
Comment 21: Line 554: “One of female adult in the experiment lays more than 20 eggs at one time and formed an egg mass” · “One of the adult females in the experiment” · Was this observation based on a single individual or multiple females? If multiple, was 20+ eggs the minimum, average, or maximum number laid in a single event? |
||
|
Response 21: Agree. We have not recorded in detail the number of male and female individuals. We have provided as much information as possible. A lot of biological information still needs to be observed.
“We recorded the life history of Rhynocoris fuscipes (Fabricius, 1787) based on about 20 live specimens collected from Zhaotong, Yunnan, China. One of the adult females in the experiment lays more than 20 eggs at one time and formed an egg mass (Figure 10A). ”
|
||
|
Comment 22: DISCUSSION Line 605: “…, R. costalis and R. fuscipes, widely distributed in Oriental Region, and are common” · “…, R. costalis and R. fuscipes, are widely distributed in the Oriental Region and are common” |
||
|
Response 22: Agree. “The two morphological similar species, R. costalis and R. fuscipes, are widely distributed in Oriental Region, and are common predatory natural...”
|
||
|
Comment 23: Line 612: “had a important, profound” “had an important” or “had a profound” Redundant, choose between important and profound |
||
|
Response 23: Agree. “Especially “A Handbook for the Determination of the Chinese Hemiptera-Heteroptera (II)” [3] has had a profound influence on taxonomic study on the sub-order Heteroptera from China. ”
|
||
|
Comment 24: Line 613: “In the book, Hsiao et al. in 1981 describe the other species R. costalis” · “In 1981, R. costalis was described, but…[3]” or “Hsiao et al. described the other species of R. costalis, but…[3]” · Not sure I understand what is meant by “the other species.” Line 614: “From then until today,” · “Since then” |
||
|
Response 24: Revised. Misidentification. “In the book, Hsiao et al. in 1981 provided description of the species R. costalis, but the given Latin name was R. fuscipes”
|
||
|
Comment 25: Line 620–623: “Over the past more than two hundred years, some of distribution information of the two widespread species cannot be verified, so the distribution records of the two species in the literature are both true and false, but the distribution information of examined specimen in the present study are true and accurate. This sentence could use some rephrasing to clarify, particularly, how did we end up with records that are both true and false and why this study is an improvement and more reliable than past sources. For example, “For over two hundred years, distribution records for both widespread species have been inconsistent/misrepresented(??) and unverified, leading to accurate and inaccurate reports in the literature. The present study provides accurate/verified distribution data based on carefully examined and identified species.” Line 684: “there are still numerous issues to be further studued” · Great job including limitations and directions for future work in the discussion. Word choice: “issues”. Consider using another word that better reflects ongoing areas of investigation rather than problems. E.g., replace it with aspects, topics, questions, etc. |
||
|
Response 25: Agree. “For over two hundred years, distribution records for both widespread species have been inconsistent and unverified, leading to accurate and inaccurate reports in the literature. The present study provides accurate distribution data based on carefully examined and identified species.”
|
||
|
Comment 26: FIGURES The labeled genitalia figures look great and are very helpful! Line 273: Figure 3. (C–E) paramere Line 468 Figure 8. (C–E) paramere Left or right paramere or both parameres? |
||
|
Response 26: Agree. Left and right paremeres have been labeled in Figures 3, 6, 8.
|
||
|
4. Response to Comments on the Quality of English Language |
||
|
Point 1: The English could be improved to more clearly express the research. The English has improved, but now the new parts of the Discussion section could use some attention. Some suggestions are provided in the attached file. |
||
|
Response 1: We try to revise the MS. Thank you for your suggestions. |
||
|
5. Additional clarifications |
||
|
|
||